# Core genes involved in the regulation of acute lung injury and their association with COVID-19 and tumor progression: A bioinformatics and experimental study

**Aleksandra V. Sen'kova**[ID]*[◉], **Innokenty A. Savin**[ID][◉], **Evgenyi V. Brenner, Marina A. Zenkova, Andrey V. Markov**

Laboratory of Nucleic Acids Biochemistry, Institute of Chemical Biology and Fundamental Medicine, Siberian Branch of the Russian Academy of Sciences, Novosibirsk, Russia

◉ These authors contributed equally to this work.
* alsenko@mail.ru

**Data Availability Statement:** All GSE files are available from the GEO database (accession

## Abstract

Acute lung injury (ALI) is a specific form of lung damage caused by different infectious and non-infectious agents, including SARS-CoV-2, leading to severe respiratory and systemic inflammation. To gain deeper insight into the molecular mechanisms behind ALI and to identify core elements of the regulatory network associated with this pathology, key genes involved in the regulation of the acute lung inflammatory response (*Il6*, *Ccl2*, *Cat*, *Serpine1*, *Eln*, *Timp1*, *Ptx3*, *Socs3*) were revealed using comprehensive bioinformatics analysis of whole-genome microarray datasets, functional annotation of differentially expressed genes (DEGs), reconstruction of protein-protein interaction networks and text mining. The bioinformatics data were validated using a murine model of LPS-induced ALI; changes in the gene expression patterns were assessed during ALI progression and prevention by anti-inflammatory therapy with dexamethasone and the semisynthetic triterpenoid soloxolone methyl (SM), two agents with different mechanisms of action. Analysis showed that 7 of 8 revealed ALI-related genes were susceptible to LPS challenge (up-regulation: *Il6*, *Ccl2*, *Cat*, *Serpine1*, *Eln*, *Timp1*, *Socs3*; down-regulation: *Cat*) and their expression was reversed by the pre-treatment of mice with both anti-inflammatory agents. Furthermore, ALI-associated nodal genes were analysed with respect to SARS-CoV-2 infection and lung cancers. The overlap with DEGs identified in postmortem lung tissues from COVID-19 patients revealed genes (*Saa1*, *Rsad2*, *Ifi44*, *Rtp4*, *Mmp8*) that (a) showed a high degree centrality in the COVID-19-related regulatory network, (b) were up-regulated in murine lungs after LPS administration, and (c) were susceptible to anti-inflammatory therapy. Analysis of ALI-associated key genes using The Cancer Genome Atlas showed their correlation with poor survival in patients with lung neoplasias (*Ptx3*, *Timp1*, *Serpine1*, *Plaur*). Taken together, a number of key genes playing a core function in the regulation of lung inflammation were found, which can serve both as promising therapeutic targets and molecular markers to control lung ailments, including COVID-19-associated ALI.

numbers GSE58654, GSE80011, GSE130936, GSE94522, GSE21802, GSE20346, GSE40012, GSE76293).

**Funding:** This research was supported by the Russian Science Foundation (grant #19-74-30011) and the Russian State funded budget projects of ICBFM #0245-2021-0004. The funders had no role in study design, data collection and analysis, decision to publish, or preparation of the manuscript.

**Competing interests:** The authors declare no conflicts of interest.

## Introduction

Acute lung injury (ALI), clinically followed by the acute respiratory distress syndrome (ARDS), is a specific form of lung injury characterised by diffuse alveolar damage, non-cardiogenic pulmonary oedema, pulmonary and systemic neutrophil-associated inflammation, finally resulting in respiratory failure and hypoxemia [1–4]. Annually, ARDS affects more than 3 million patients worldwide with mortality ranging from 35% to 46% [5, 6]. Over the past year, morbidity and mortality from this pathology have increased significantly with the main contribution of coronavirus disease 2019 (COVID-19) [7, 8].

ALI can be caused by different phlogogens and irritants, such as bacterial and viral pathogens [9–12], chemicals such as chlorine and phosgene [13, 14], and industrial aerosols [15]. However, at the present time, the most common causative agent of ALI is severe acute respiratory syndrome coronavirus 2 (SARS-CoV-2) infection, resulting in pneumonia and ARDS [16].

In animal studies, exposure to different chemical agents can induce a unique mix of physiologic derangements ultimately leading to pathology, similar to the acute phase of ARDS. The most prevalent methods of ALI induction are bacterial lipopolysaccharide (LPS) [17, 18] or bleomycin [19] injection, hyperoxic injury [20], and influenza virus infection [21]. LPS, a major component of the outer membrane of Gram-negative bacteria and a prominent inducer of local and systemic inflammatory responses, is closely related to lung injury and has often been employed to induce pulmonary inflammation in ALI *in vivo* models [22–24].

Despite the diverse origin, the pathogenesis of ALI is conventional and is associated with the development of an inflammatory cascade reaction in response to lung insult, which leads to increased pulmonary vascular permeability, diffuse alveolar damage and, as a result, respiratory insufficiency [25–27]. Briefly, in response to injury, inflammatory lung macrophages develop a pro-inflammatory (M1) phenotype, releasing pro-inflammatory cytokines (TNF-α, IL-6, IL-1) and chemokines (IL-8, CCL7, CCL2), and promoting further accumulation of monocytes and neutrophils in the alveolar space [28]. Neutrophils release additional inflammatory mediators, reactive oxygen species (ROS), and proteinases, which degrade the basement membrane and epithelial-endothelial barrier. Inactivation and loss of surfactant result in alveolar collapse. Moreover, the pathogenesis of ALI and ARDS also involves multiple factors such as coagulation/fibrinolysis imbalance, apoptosis, and redox dysfunction [27]. The combination of these pathologic processes results in increased dead space ventilation and intrapulmonary shunting of blood, culminating in hypoxemia and ultimately respiratory failure [13].

Pathomorphological changes in damaged lungs during ALI development are also unified and generally represented by the neutrophil- and lymphocyte-associated inflammatory infiltration and diffuse alveolar damage displayed by alveolar and interstitial oedema and hyaline membrane formation in the exudative phase, and by fibroblast/myofibroblast proliferation, extracellular matrix deposition, and intra-alveolar fibrin accumulation in the proliferative phase [26, 29, 30]. The alveolar and epithelial structure also undergo non-specific alterations; the most frequently occurring are squamous metaplasia, intra-alveolar haemorrhage, desquamation, and type 2 pneumocyte hyperplasia [29]. It should be noted that the immunopathological and histological landscape of SARS-CoV-2-mediated lung injury has the same morphological characteristics and pathogenetic features [29, 31]. Thus, these inflammatory, destructive, and dyscirculatory changes in the lung tissue are common for both infectious and non-infectious pulmonary diseases of variable origin and cannot be considered pathognomonic or highly specific for COVID-19 [32].

The quick and early prediction of a severe disease course and complications of ALI/ARDS through specific biological markers would prevent many deaths, including those from

COVID-19-associated pulmonary failure. The use of these markers could allow for closer clinical monitoring and supportive treatment to prevent a poor prognosis. For example, the clinical markers that can diagnose the severity of tissue damage in SARS-CoV-2 infection include serum biochemistry and haematological parameters of the cytokine storm: ferritin, D-dimer, lactate dehydrogenase, C-reactive protein, ALT, cytokines (IL-1β, IL-6, TNF-α), the neutrophil/lymphocyte ratio, and the erythrocyte sedimentation rate [33]. Moreover, a number of studies have shown that the level of some cytokines and chemokines, for example, CXCL13, in biological fluids (serum, plasma, bronchoalveolar lavage fluid) and lung tissues can serve as a biomarker of a lethal course of both SARS-CoV-2 infection [34], and other severe pulmonary pathologies (non-small lung cancer, idiopathic pulmonary fibrosis), which can be fatal [35–38]. Similar correlations were shown for a number of other pro-inflammatory cytokines and chemokines (IL-6, CCL2), whose tissue and serum expression levels are associated with poor survival in patients with lung cancer and pulmonary fibrosis [39–45].

Here, in order to reveal the key molecular mechanisms underlying ALI and to identify the core master regulators involved in the development of lung inflammation of various aetiologies, bioinformatics analysis of ALI-related whole-genome microarray datasets, functional enrichment analysis of differentially expressed genes, protein-protein interaction network reconstruction, and data mining analysis were performed. Bioinformatics analysis data were validated using a model of LPS-induced ALI in mice and treatment with anti-inflammatory compounds with different mechanisms of action: the glucocorticosteroid dexamethasone and the semisynthetic derivative of 18βH-glycyrrhetinic acid soloxolone methyl, the marked anti-inflammatory activity of which has been shown by us previously [46, 47]. Furthermore, the role of potential master regulators in the development of ALI was analysed with respect to SARS-CoV-2 infection and lung cancers. We have revealed gene expression patterns common to these pathologies and identified key genes that can serve both as molecular markers and potential targets for the treatment of inflammatory lung ailments, including ALI associated with COVID-19.

## Materials and methods

### GEO dataset analysis

The gene expression profiles of GSE58654 (3 PBS treated mice, 3 mice with hyperoxia lung injury), GSE130936 (3 saline treated mice, 3 LPS treated mice), GSE80011 (3 healthy mice, 3 influenza-virus infected mice), GSE94522 (3 healthy mice, 3 bleomycin treated mice), GSE21802 (4 healthy samples; 6 influenza A-induced pneumonia samples without mechanical ventilation), GSE20346 (18 healthy samples; 6 bacterial pneumonia samples; 4 influenza A-induced pneumonia samples); GSE40012 (18 healthy samples; 16 bacterial pneumonia samples; 8 influenza A-induced pneumonia samples); GSE76293 (10 healthy samples; 12 acute respiratory distress syndrome (ARDS) samples) were acquired from the Gene Expression Omnibus (http://www.ncbi.nlm.nih.gov/geo/) database. All of the analysed datasets concern processes associated with acute lung injury in mice at a time point earlier than 24 h after the induction and in humans at day 1 after admission. The characteristics of used datasets are presented in S1 Table.

The identification of differentially expressed genes (DEGs) between healthy mice and mice with ALI was performed using GEO2R, a web service that allows the comparison of two or more datasets in GEO series for DEGs identification across experimental conditions [48]. The adjusted $p$-values were applied to correct the false positive results by the default Benjamin-Hochberg false discovery rate method. The adjusted $p < 0.05$ and |Fold Change| $> 1.5$ were viewed as the cutoff values. A Venn diagram analysis of revealed DEGs was performed using Venny v. 2.0 tool (https://bioinfogp.cnb.cs/tools/venny/).

## Functional enrichment analysis

To elucidate the pathways and biological processes affected by ALI development, we performed functional enrichment analysis of DEGs using ClueGo v.2.5.1 plugin in Cytoscape v 3.8.1 [49]. DEGs were mapped on the latest update of Gene Ontology (biological processes), Kyoto Encyclopedia of Genes and Genomes (KEGG), REACTOME and Wikipathways. Only terms with $p \leq 0.05$ were included in the analysis.

## PPI network reconstruction

The protein-protein interactions (PPIs) were predicted based on data deposited in the Search Tool for the Retrieval of Interacting Genes/Genomes (STRING) database, with a confidence score > 0.7. The reconstructed protein-protein pairs included functional relationships of proteins from five sources: published high-throughput experiments, genomic context prediction, co-expression, automated text mining, and PPI deposited in other databases. Reconstructed PPI networks were visualised as undirected or hierarchical networks using Cytoscape. To identify hub proteins most interconnected with their neighbours in the PPI network node, degree scores were calculated using NetworkAnalyzer plugin [50], and degree centrality values of the nodes were visualised as heatmaps using the Morpheus tool (https://software.broadinstitute.org/morpheus/). A cut-off criterion to identify hub genes/proteins is degree of node more than 10 within the analysed network.

## Data mining analysis

To analyse the co-occurrence of genes of interest and keywords associated with lung pathology in scientific texts deposited in MEDLINE database, a data-mining analysis of scientific literature was performed using GenClip3 web-service [51]. The list of identified DEGs common for all five analysed ALI-associated datasets was uploaded into GenClip3 and a search of co-occurrence of identified DEGs with the following keywords was performed: COVID-19, pneumonia, lung inflammation, influenza, hyperoxia, LPS AND lung, acute lung injury OR ALI, bleomycin AND lung, ARDS.

## Survival analysis of DEGs

In order to analyse the involvement of identified DEGs in the progression of lung adenocarcinoma (LUAD) and lung squamous cell carcinoma (LUSC), analysis of survival rates and their correlations with the expression of studied genes was performed using The Cancer Genome Atlas (TCGA) clinical data for patients with LUAD and LUSC. The visualisation of obtained data was carried out using Circos Table Viewer (http://mkweb.bcgsc.ca/tableviewer/). The width of ribbons corresponds with Log-rank $p$-value, with wider ribbons indicating the most significant correlations. Kaplan-Meier survival curves for LUAD and LUSC depending on mRNA expression level were constructed based on the TCGA data by using OncoLnc tool (http://www.oncolnc.org/). Kaplan-Meier survival curves for lung cancers depending on protein expression level were constructed based on The Human Protein Atlas data (https://www.proteinatlas.org/).

## Mice

Female 6-8-week-old Balb/C mice with an average weight of 20–22 g were obtained from the Vivarium of Institute of Chemical Biology and Fundamental Medicine SB RAS (Novosibirsk, Russia). Mice were housed in plastic cages (6 animals per cage) under normal daylight conditions. Water and food were provided ad libitum. Experiments were carried out in accordance

with the European Communities Council Directive 86/609/CEE. The experimental protocols were approved by the Committee on the Ethics of Animal Experiments at the Institute of Cytology and Genetics SB RAS (Novosibirsk, Russia) (protocol No. 56 from August 10, 2019).

## In vivo model of Acute Lung Injury (ALI)

For the *in vivo* study, mice were randomly divided into five groups with 6 mice in each group. ALI was induced by intranasal (i.n.) instillation of LPS (055:B5, Sigma-Aldrich, USA) 10 μg per mice under isoflurane anaesthesia.

As compounds with demonstrated anti-inflammatory activity, the semisynthetic derivative of 18βH-glycyrrhetinic acid soloxolone methyl (SM) and the glucocorticosteroid dexamethasone (Dex) were used. SM was synthesised, characterised, and kindly provided by Dr Oksana V. Salomatina and Prof Nariman F. Salakhutdinov (N.N. Vorozhtsov Novosibirsk Institute of Organic Chemistry SB RAS, Novosibirsk, Russia) [47, 52, 53].

SM (10 mg/kg) and Dex (1 mg/kg) diluted in DMSO: sesame oil (1:9) were administered to mice via gastric gavage 1 h before LPS challenge. LPS-exposed animals treated with vehicle only (DMSO: sesame oil (1:9)) (hereinafter, vehicle) or without treatment (hereinafter, ALI) were used as controls. Mice were sacrificed 24 h after induction of inflammation. The lung tissues and bronchoalveolar lavage fluid were collected for subsequent analysis.

## Bronchoalveolar lavage (BAL) fluid analysis

The lungs were lavaged with 1 mL ice-cold phosphate buffered saline (PBS) twice in all groups. The recovery ratio of the fluid was about 80–90%. The collected BAL fluid was centrifuged at 1500 rpm for 10 min at 4˚C and the supernatant was collected for subsequent protein study. The cell pellets were resuspended in 50 μL of PBS, and total leukocyte counts were performed with a Neubauer chamber by optical microscopy after diluting in Türk solution (1:20). To determine the differential leukocyte counts, bronchoalveolar cells were centrifuged and placed onto slides, stained with azur-eosin by the Romanovsky-Giemsa method, and examined by optical microscopy. The results were expressed as the number of total leukocytes ($\times 10^4$ cells/mL) and the percentages of subpopulations of neutrophils, lymphocytes, and monocytes (%).

## ELISA

BAL fluid supernatants were analysed for the pro-inflammatory cytokine TNF-α by ELISA (Thermo Scientific, Rockford, IL, USA) according to the manufacturer's instructions. Briefly, 50 μL of supernatant and 50 μL of sample diluent were placed to wells with immobilised monoclonal antibodies against TNF-α, filled with 50 μL of biotinylated anti-TNF-α antibody solution and incubated for 2 h at room temperature and 400 rpm in thermostatic shaker ST-3M (ELMI Ltd, Riga, Latvia). Next, the wells were sequentially filled with 100 μL of horseradish peroxidase labelled streptavidin (HRP-streptavidin) solution, 100 μL of 3,3',5,5'-tetamethyl-benzidine (TMB) solution and incubated for 1 h at room temperature and 400 rpm (for HPR-streptavidin) and 30 min at room temperature without direct light (for TMB). Before each addition, wells were washed five times with wash buffer. Stop solution was added to each well and the absorbance was measured at 450 nm using a Multiscan RC plate reader (Thermo Lab-systems, Vantaa, Finland).

## Histology and immunohistochemistry

For the histological study, lung specimens were fixed in 10% neutral-buffered formalin (BioVitrum, Moscow, Russia), dehydrated in ascending ethanols and xylols and embedded in

HISTOMIX paraffin (BioVitrum, Russia). The paraffin sections (5 μm) were sliced on a Microm HM 355S microtome (Thermo Fisher Scientific, Waltham, MA, USA) and stained with haematoxylin and eosin.

The expression of neutral mucins in lung tissue was determined using periodic acid-Schiff (PAS) staining. The tissue sections were deparaffinised, rehydrated, and stained with Schiff's reagent (BioVitrum, Russia) according to a standard protocol and counterstained with haematoxylin.

For the immunohistochemical study, lung sections (3–4 μm) were deparaffinised and rehydrated. Antigen retrieval was carried out after exposure in a microwave oven at 700 W. The samples were incubated with the anti-TNF-α specific monoclonal antibodies (ab212899, Abcam, Cambridge, MA, USA) according to the manufacturer's protocol. Then, the sections were incubated with secondary horseradish peroxidase (HPR)-conjugated antibodies, exposed to the 3,3'-diaminobenzidine (DAB) substrate (Rabbit Specific HRP/DAB (ABC) Detection IHC Kit, ab 64261, Abcam, USA), and stained with Mayer's haematoxylin.

All the images were examined and scanned using Axiostar Plus microscope equipped with Axiocam MRc5 digital camera (Zeiss, Oberkochen, Germany) at magnifications of ×100 and ×400.

## Quantitative real-time PCR (qRT-PCR)

Total RNA was isolated from lungs of experimental animals using TRIzol Reagent (Ambion, Austin, TX, USA) according to the manufacturer's recommendation. Briefly, lung tissue was collected in 1.5 mL capped tubes, filled with 1 g of lysing matrix D (MP Biomedicals, Irvine, CA, USA) and 1 mL of TRIzol reagent, then homogenised using a FastPrep-24TM 5G homogeniser (MP Biomedicals, Irvine, CA, USA), using QuickPrep 24 adapter. The homogenisation was performed at 6.0 m/s for 40 s. After homogenisation, the content of the tubes was transferred to the new 1.5 mL tubes without lysing matrix. Total RNA extraction was performed according to the TRIzol reagent protocol.

The first-strand of cDNA was synthesised from total RNA in 100 μL of reaction mixture containing 2.5 μg total RNA, 20 μL of 5× RT buffer (Biolabmix, Novosibirsk, Russia), 250 U M-MuLV-RH revertase (Biolabmix, Novosibirsk, Russia), and 100 μM of random hexaprimers (5'-NNNNNN-3') in a volume of 100 μL. Reverse transcription was performed at 25°C for 10 min followed by incubation at 42°C for 60 min. Finally, reverse transcriptase was terminated at 70°C for 10 min.

In the case of *Il-6*, *Ccl2*, *Cat*, *Serpine1*, *Eln*, *Timp1*, *Ptx3*, *Socs3*, and *HPRT*, amplification of cDNA was performed in a 25 μL PCR reaction mixture containing 5 μL of cDNA, 12.5 μL of HS-qPCR (2×) master mix (Biolabmix, Novosibirsk, Russia), 0.25 μM each of forward and reverse primers to *HPRT* and *HPRT* specific ROX-labelled probe; 0.25 μM each of forward and reverse gene-specific primers and FAM-labelled probe (Table 1). Amplification was performed as follows: (1) 94°C, 2 min; (2) 94°C, 10 s; 60°C, 30 s (50 cycles). The relative level of gene expression was normalised to the level of *HPRT* according to the ΔΔCt method.

In the case of *TNF-α*, *Il-1β*, and *GAPDH*, amplification of cDNA was performed in a 25 μL PCR reaction mixture containing 5 μL of cDNA, 12.5 μL of HS-qPCR SYBR Blue (2×) master mix (Biolabmix, Novosibirsk, Russia), 0.25 μM each of forward and reverse primers to *GAPDH*; 0.25 μM of each forward and reverse gene-specific primers (Table 1). Amplification was performed as follows: (1) 95°C, 5 min; (2) 95°C, 10 s; (3) 51°C 30 s; (4) 72°C, 30 s (40 cycles); (5) 65°C to 95°C in 0.5°C increments every 5 s. The relative level of gene expression was normalised to the level of *GAPDH* according to the ΔΔCt method.

The relative level of gene expression was determined with a CFX96™ Real-Time system (C1000 Touch™, USA).

**Table 1. The primers used in this study.**

| Gene | Type | Sequence |
|---|---|---|
| Il-6 | Forward | 5'-AAACCGCTATGAAGTTCCTCTC-3' |
| | Probe | 5'-((5,6)-FAM)-TTGTCACCAGCATCAGTCCCAAGA-BHQ1-3' |
| | Reverse | 5'-GTGGTATCCTCTGTGAAGTCTC-3' |
| Ccl2 | Forward | 5'-TCCACTACCTTTTCCACAACC-3' |
| | Probe | 5'- ((5,6)-FAM)-AAGGCATCACAGTCCGAGTCACAC-BHQ1-3' |
| | Reverse | 5'-GGATCCACACCTTGCATTTAAG-3' |
| Catalase | Forward | 5'-TTCCATCCTTTATCCATAGCCAG-3' |
| | Probe | 5'-((5,6)-FAM)-ACTCCAGAAGTCCCAGACCATGTCA-BHQ1-3' |
| | Reverse | 5'-GAATCCCTCGGTCACTGAAC-3' |
| Serpine1 | Forward | 5'-ACACACAGCCAACCACAG-3' |
| | Probe | 5'—((5,6)-FAM)-ACAGCCAACAAGAGCCAATCACAAG-BHQ1-3' |
| | Reverse | 5'-TCCCAGAGACCAGAACCAG-3' |
| Elastin | Forward | 5'-CTTATAAAGCTGCCGCCAAA-3' |
| | Probe | 5'- ((5,6)-FAM)-ACTCCGCCAACTCCAACACCA-BHQ1-3' |
| | Reverse | 5'-ACTCCACCAACTCCAACAC-3' |
| Timp1 | Forward | 5'-CTCAAAGACCTATAGTGCTGGC-3' |
| | Probe | 5'-((5,6)-FAM)-ACTCACTGTTTGTGGACGGATCAGG-BHQ1-3' |
| | Reverse | 5'-CAAAGTGACGGCTCTGGTAG-3' |
| Ptx3 | Forward | 5'-AGCAAATTTCGCCTCTCCAG-3' |
| | Probe | 5'- ((5,6)-FAM)-AAGCAGGATCGCAGGGAGGTG-BHQ1-3' |
| | Reverse | 5'-GTCCATTGTCTATTTCGTTGTCC-3' |
| Socs3 | Forward | 5'-CCTATGAGAAAGTGACCCAGC-3' |
| | Probe | 5'- ((5,6)-FAM)-CCCCTCTGACCCTTTTGCTCCTT-BHQ1-3' |
| | Reverse | 5'-TTTGTGCTTGTGCCATGTG-3' |
| Mmp8 | Forward | 5'-CATATCTCTGTTCTGGCCCTTC-3' |
| | Probe | 5'- ((5,6)-FAM)-TACCCAACGGTCTTCAGGCTGC-BHQ1-3' |
| | Reverse | 5'-CAGGTCATAGCCACTTAGAGC-3' |
| Rsad2 | Forward | 5'-TGGATGTTGGCGTGGAAG-3' |
| | Probe | 5'- ((5,6)-FAM)-TCTGAAGCGTGGCGGAAAGTATGT-BHQ1-3' |
| | Reverse | 5'-CTGTAGCTGGTCGGAGTTTC-3' |
| Rtp4 | Forward | 5'-GTTCCCCGATGACTTCAGTAC-3' |
| | Probe | 5'- ((5,6)-FAM)-TTGGCAGGTTCCAGTGTTCCAGAT-BHQ1-3' |
| | Reverse | 5'-CTGAGCAGAGGTCCAACTTC-3' |
| Ifi44 | Forward | 5'-GAACTATACCCATGACCCACTG-3' |
| | Probe | 5'- ((5,6)-FAM)-CCACCAGCTCAGAAGAGTGCATTTCA-BHQ1-3' |
| | Reverse | 5'-GTAATCAGATCCAGGCTATCCAC-3' |
| Saa1 | Forward | 5'-CAGGATGAAGCTACTCACCAG-3' |
| | Probe | 5'- ((5,6)-FAM)-CATTTGTTCACGAGGCTTTCCAAGGG-BHQ1-3' |
| | Reverse | 5'-CTTCATGTCAGTGTAGGCTCG-3' |
| HPRT | Forward | 5'-CCCCAAAATGGTTAAGGTTGC-3' |
| | Probe | 5'- ((5,6)-ROX)-CTTGCTGGTGAAAAGGACCT-BHQ2-3' |
| | Reverse | 5'-AACAAAGTCTGGCCTGTATCC-3' |
| TNF-α | Forward | 5'-TCAGCCTCTTCTCATTCCTG-3' |
| | Reverse | 5'-TGAAGAGAACCTGGGAGTAG-3' |
| Il-1β | Forward | 5'-TGCAGAGTTCCCCAACTGGTACAT-3' |
| | Reverse | 5'-GTGCTGCCTAATGTCCCCTTGAAT-3' |
| GAPDH | Forward | 5'-AAGAGAGGCCCTATCCCAAC-3' |
| | Reverse | 5'-GCAGCGAACTTTATTGATGG-3' |

## Statistical analysis

The data are expressed as the mean ± SD. The statistical analysis was performed using the two-tailed unpaired t-test; $p$-values of less than 0.05 were considered statistically significant.

## Results

### Identification of key genes involved in the development of ALI-associated lung inflammation

In order to identify the potential key genes, regulating the development of acute inflammation in lungs as one of the main manifestations of ALI caused by the different damaging factors, the changes in gene expression between inflamed and normal lung tissues were assessed by re-analysis of cDNA microarray data from GSE58654 (hyperoxia), GSE80011 (influenza), GSE130936 (LPS), and GSE94522 (bleomycin) datasets using the GEO2R tool (Fig 1, S1 Table). Datasets containing experimental data with ALI inducers of diverse origin, but resulting in similar consequences for the organism, were analysed simultaneously in order to identify core genes common for lung pathology of various aetiologies.

Venn diagrams analysis of the revealed DEGs (fold change > 1.5, $p < 0.05$) are displayed in Fig 1A. We identified in total 58 DEGs, common for all analysed GEO datasets. Functional analysis of the identified DEGs revealed high enrichment of terms, associated predominately with the regulation of cytokine production, type 2 immune response, cellular response to interleukin-1, lung fibrosis, and, to a lesser extent, with vitamin B6 metabolism, cellular response to copper ion, and nitric oxide synthesis (Fig 1B).

In the next step, we clustered gene expression data using the Euclidian distances method. Identified DEGs were clustered into two main clades, containing 48 up-regulated and 10 down-regulated DEGs (Fig 1C). Interestingly, despite the variable nature and mechanisms of ALI induction, gene expression profiles of identified DEGs were largely the same between the analysed datasets. Furthermore, in order to identify the potential regulatory role of overlapping DEGs in lung inflammation, PPI networks were reconstructed for all analysed GEO datasets using the Search Tool for the Retrieval of Interacting Genes/Proteins (STRING) database, retrieving relatively high confidence protein interactions (confidence score: 0.7), followed by the identification of the degree centrality of each of the 58 overlapping ALI-associated key DEGs within the reconstructed networks (Fig 1D).

The top 10 key nodes, i.e. the most interconnected within the ALI-related PPI networks, mainly include (a) pro-inflammatory cytokines and chemokines (*Il-6*, *Cxcl13*, *Ccl2*), taking part in a wide spectrum of inflammatory diseases, including viral and bacterial infections [55–58] and (b) regulators of interactions of cells with other cells or extracellular matrix (ECM), including *Timp1*, the inhibitor of matrix metalloproteinases responsible for ECM degradation, which promotes cell proliferation and exacerbates lung inflammation and fibrosis [59, 60], *Adam8*, a membrane protein responsible for cell-to-cell and cell-to-matrix interactions, which drives and controls acute and chronic lung inflammation [61], *Thbs1*, an adhesive glycoprotein that also mediates the interaction of cells with other cells and the ECM and the expression of which has been found to be increased during inflammation [62], as well as *Serpina3n* and *Serpine1*, serine protease inhibitors involved in ECM remodelling and related to inflammation [63, 64]. Additionally, the top 10 ALI-associated key nodes also included *Socs3*, a suppressor of cytokine signalling and negative regulator of JAK1/STAT3 signalling axis [65], and *Rtp4*, a type I interferon-induced gene involved in immune response and viral defence [66]. Interestingly, the listed top 10 hub genes are tightly interconnected with the rodent inflammatome, a list of DEGs identified previously by Wang et al. in 11 independent rodent models of

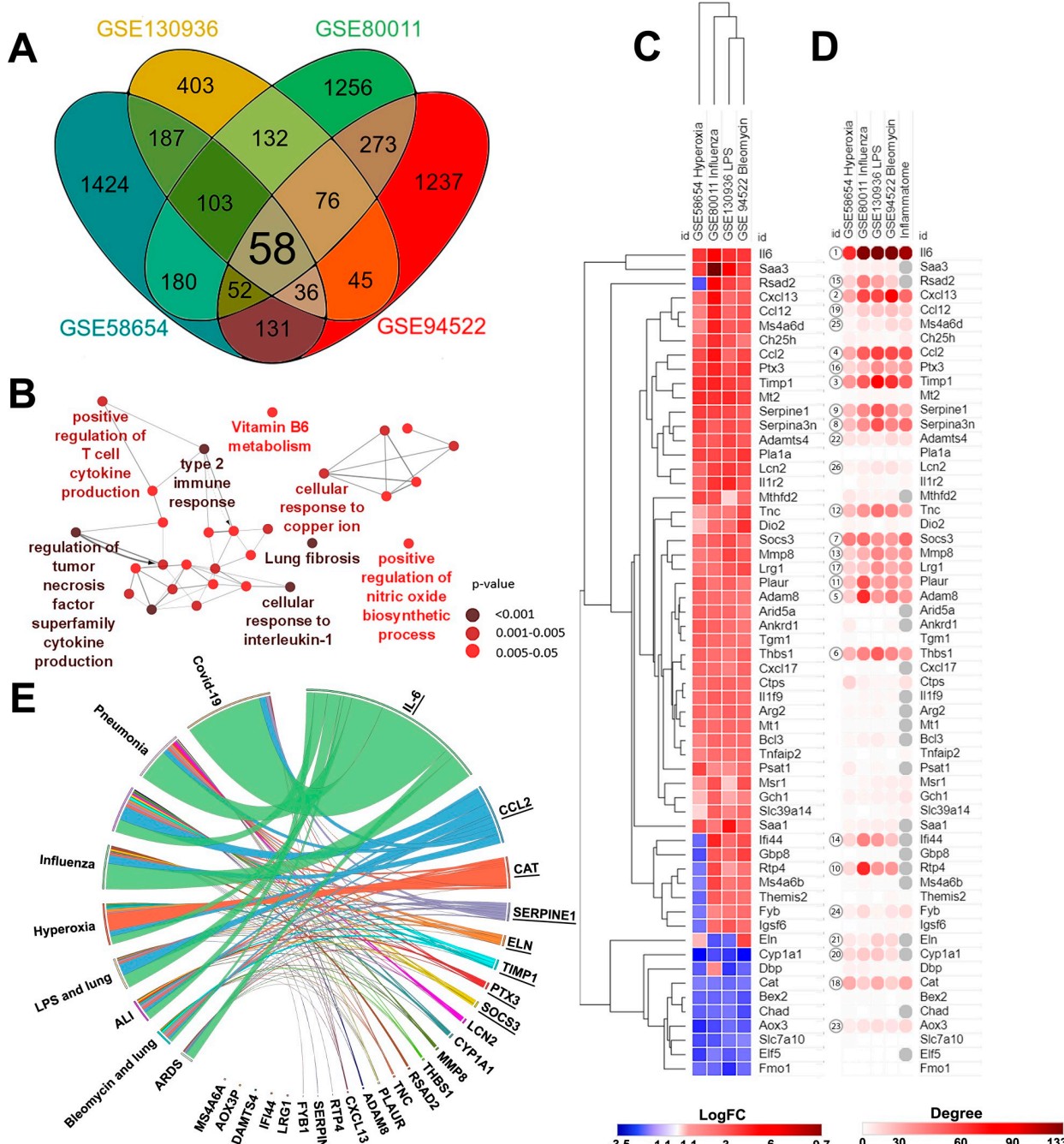

**Fig 1. Key genes involved in the development of Acute Lung Injury (ALI): Results of bioinformatics analysis.** (**A**) Venn diagram illustrating overlap between differentially expressed genes (DEGs) identified by the analysis of GSE80011, GSE130936, GSE58654, and GSE94522 datasets of ALI caused by different agents. (**B**) Functional analysis of overlapping DEGs identified in different GSEs. Enrichment for Gene Ontology (biological processes), KEGG, REACTOME, and Wikipathways terms were performed using the ClueGo plugin in Cytoscape. The labels of the most significant terms are shown. Colours of nodes represent the term enrichment significance. Only pathways with $p < 0.05$ after the Bonferroni step-down correction for multiple testing were included in the networks. (**C**) Heat map showing expression levels of DEGs in different GSEs. Heat map construction and hierarchical clustering (Euclidean distances) were performed using Morpheus. LogFC = $Log_2$ (fold change). (**D**) Heat map demonstrating the interconnection of DEGs in a protein-protein interaction network (PPI) reconstructed for each ALI dataset using the STRING database (confidence score $\geq$ 0.7, maximal number of interactors = 0) in Cytoscape. Degree: number of interactions between a DEG and its partners. Inflammatome: number of interactions between DEGs and partners in the rodent inflammatome, constructed based on transcriptomic data published by Wang et al. [54]. (**E**) Co-occurrence of identified DEGs with relevant keywords in the scientific literature deposited in the MEDLINE database. Analysis was performed using the GenClip3 web service. Data were visualised via Circos.

inflammatory diseases [54] (Fig 1D), which clearly showed their key regulatory roles in inflammation.

As depicted in Fig 1D, other DEGs displaying a high degree centrality and forming top-25 key nodes include modulators of ECM homeostasis (urokinase-type plasminogen activator (*Plaur*), matrix metalloproteinase 8 (*Mmp8*), elastin (*Eln*), ADAM metallopeptidase with thrombospondin type 1 motif 4 (*Adamts4*), tenascin C (*Tnc*)), immune response molecules (interferon induced protein 44 (*Ifi44*), viperin (*Rsad2*), pentraxin 3 (*Ptx3*), lipocalin 2 (*Lcn2*)), signal transduction regulators (leucine rich alpha 2 glycoprotein 1 (*Lrg1*), adhesion and degranulation promoting adaptor protein (*Fyb1*), membrane spanning 4-domains A6A (*Ms4a6a*)), as well as genes encoding the key antioxidant enzyme catalase (*Cat*), cytochrome P450 (*Cyp1a1*), and the aldehyde oxidase pseudogene (*Aox3p*).

Furthermore, in order to analyse how thoroughly these top 25 nodal genes and their protein products have been investigated in the field of pulmonary inflammation and injury, the co-occurrence of their names and lung pathology-associated keywords was analysed within the same sentences of scientific texts deposited in the MEDLINE database, using the GenCLiP3 web tool (Fig 1E). For this search, human orthologs of the top 25 nodal genes were uploaded to GenCLiP3 and the following keywords were used: COVID-19, pneumonia, lung inflammation, influenza, hyperoxia, LPS AND lung, acute lung injury OR ALI, bleomycin AND lung, ARDS. The text-mining analysis showed that only 20 of the top 25 hub genes were associated with lung injury-related processes; among them, the most interconnected genes were *IL-6* > *CCL2* > *CAT* > *SERPINE1* > *ELN* > *TIMP1* > *PTX3* > *SOCS3* (listed in descending order of their linkage with lung injury-associated keywords) (Fig 1E). Thus, in the text-mining list, both top 10 included genes (*Il-6*, *Timp1*, *Ccl2*, *Socs3*, *Serpine1*) and top 25 included genes were present: *Ptx3*, a pattern-recognition molecule, which acts as a key component of humoral innate immunity in viral infections and plays a complex regulatory role in inflammation, as well as ECM organisation and remodelling [67], *Cat*, an antioxidant enzyme with a central role in combating reactive oxygen species [68], and *Eln*, a major component of elastic fibres, an integral part of the ECM that can either increase or decrease with the development of inflammation of various aetiologies [69–72].

Given the involvement of these genes in the regulation of lung injury revealed by the text-mining approach and their nodal positions within ALI-related PPI networks, *Il-6* (rank score = 1), *Timp1* (3), *Ccl2* (4), *Socs3* (7), *Serpine1* (9), *Ptx3* (16), *Cat* (18), and *Eln* (21) were selected for validation of their involvement in ALI development on an *in vivo* model of acute lung inflammation.

## Acute Lung Injury (ALI) model

In the next step of the study, we questioned whether the revealed ALI-associated hub genes change their expression during the development and resolution of lung inflammation and whether the expression of these genes responds to anti-inflammatory therapy. In order to answer these questions, LPS-induced ALI in mice was used as an experimental model. LPS-challenged mice received anti-inflammatory therapy with the glucocorticosteroid dexamethasone (Dex) or the semisynthetic derivative of 18βH-glycyrrhetinic acid soloxolone methyl (SM), which possess anti-inflammatory action via different molecular mechanisms (Fig 2A).

Glucocorticosteroids are known to mediate their action through binding with glucocorticosteroid receptors (GR), followed by nuclear translocation and the induction of transcription of anti-inflammatory genes such as lipocortin I and calpactin [73] or binding to transcription complexes of various pro-inflammatory factors with their subsequent transcriptional repression [74]. In the case of SM, its exact anti-inflammatory mechanism of action is still unknown.

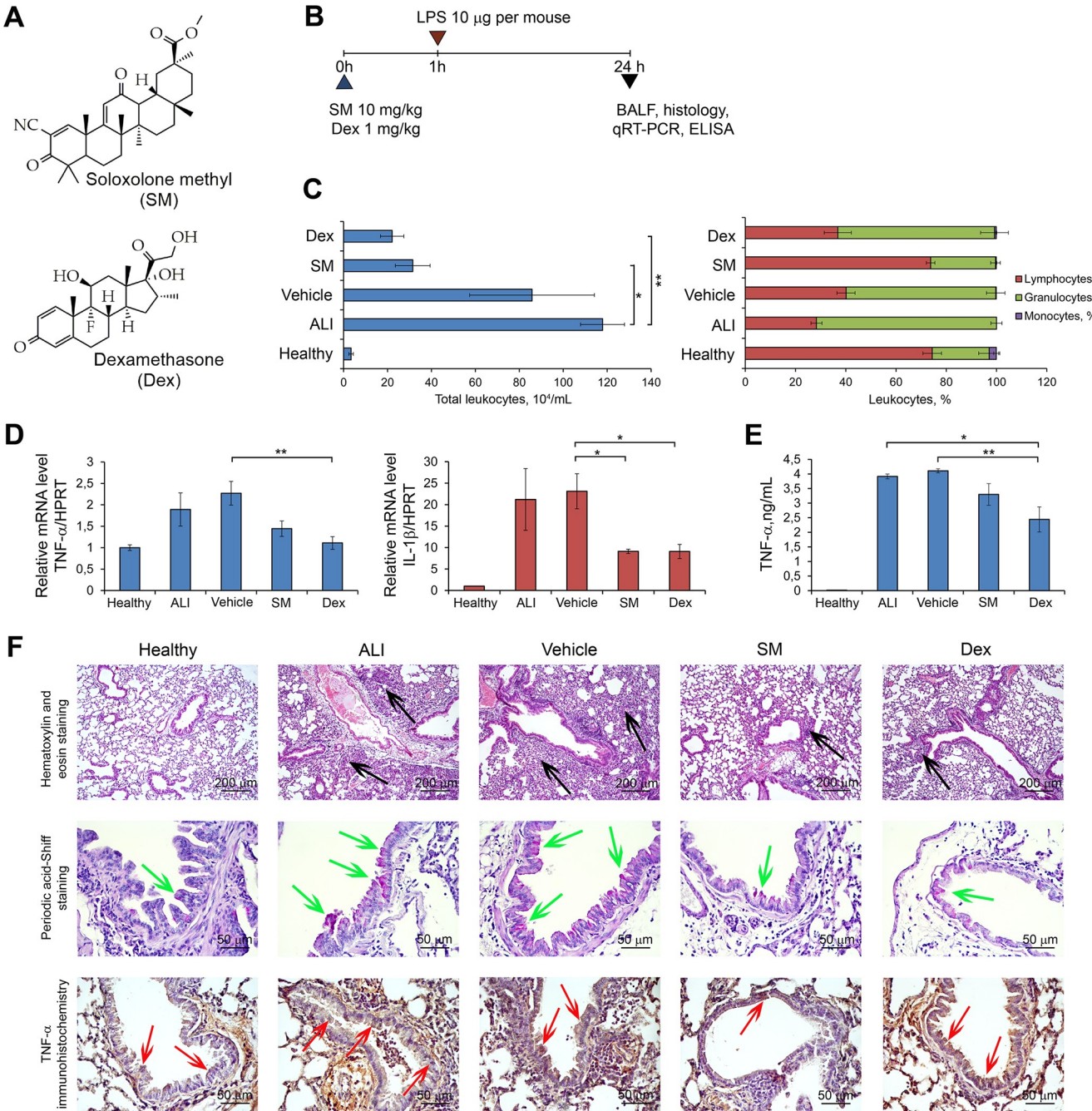

**Fig 2. Soloxolone Methyl (SM) and dexamethasone (Dex) effectively ameliorate Acute Lung Injury (ALI) *in vivo*.** (**A**) The chemical structures of SM and Dex. (**B**) The experimental setup. SM (10 mg/kg) and Dex (1 mg/kg) were administered to mice 1 h prior to LPS instillation (10 mg per mice), followed by the evaluation of ALI 24 h after the induction of lung inflammation. (**C**) Total (left) and differential (right) leukocyte counts in the bronchoalveolar (BAL) fluid of healthy and LPS-challenged mice receiving SM or Dex therapy. Total leukocyte counts were estimated using a Neubauer chamber. The distribution of leukocyte subpopulations was measured by microscopy after staining of cells with azur-eosin by Romanovsky-Giemsa. (**D, E**) The levels of pro-inflammatory cytokines (TNF-α and IL-1β) in the lung tissue measured by qRT-PCR and BAL fluid measured by ELISA. (**F**) Representative histological images of lung sections of healthy and LPS-challenged mice receiving SM or Dex therapy. Haematoxylin and eosin staining, original magnification ×100 (upper panel), periodic acid-Schiff staining, original magnification ×400 (middle panel), and immunohistochemical staining with anti-TNF-α primary antibodies, original magnification ×400 (bottom panel). The black arrows indicate inflammatory infiltration, the green arrows indicate mucus hyperproduction, and red arrows indicate TNF-α hyperexpression in lung tissue.

There is a growing amount of data showing that SM and its structural analogues are multitarget compounds, potentially exerting anti-inflammatory action through several mechanisms, such decreased NO production [53], inhibition of COX-2 expression [75–78], and suppression of the NF-κB signalling pathway [79].

As the first step, we studied in detail the development of ALI in mice after LPS induction and after anti-inflammatory therapy. The experimental setup is depicted in Fig 2B. Mice were pre-treated with SM or Dex 1 h before LPS administration (10 μg/mice). Analysis of lung injury was performed 24 h after LPS challenge. As shown in Fig 2C, the intranasal administration of LPS caused the development of acute inflammation in the respiratory system of mice, characterised by a 34.7-fold increase in the number of total leukocytes in the BAL fluid, represented predominately by granulocyte subpopulations compared with healthy animals. Evaluation of the pro-inflammatory cytokines both at the mRNA and protein levels showed 1.9- and 23.4-fold increases in the expression of TNF-α and IL-1β in the lung tissue measured by qRT-PCR and a 261.3-fold increase in the level of TNF-α in the BAL fluid measured by ELISA in LPS-stimulated mice compared with healthy animals (Fig 2D and 2E).

Histologically, the lung tissue of ALI mice was characterised by significant neutrophil-dominant inflammatory infiltration, circulatory disturbances (blood congestion, haemorrhages, interstitial and alveolar oedema) and destructive changes in the bronchial and alveolar epithelium (Fig 2F, upper panel). Moreover, reactive hyperproduction of mucus by the bronchial epithelium, which is a compensatory reaction of tissue to the damage by inflammatory cells and identified by PAS staining [80], was detected in the lungs of ALI mice (Fig 2F, middle panel). Representative images of immunohistochemical staining of lung sections with anti-TNF-α monoclonal antibodies show that healthy lungs were characterised by basal expression of TNF-α, which is in complete agreement with literature data [81]. LPS exposure significantly increased the TNF-α level in the lung tissue, especially in the bronchial epithelium, of mice without treatment compared with healthy animals.

The administration of anti-inflammatory drugs prevented the development of LPS-induced acute inflammation in the respiratory system of mice. The vehicle, used as a control for the effectiveness of anti-inflammatory therapy, did not demonstrate any positive effects on LPS-induced lung inflammation (Fig 2C–2F). ALI treatment with SM and Dex led to a 2.7- to 5.3-fold decrease in the total leukocyte number in the BAL fluid compared with the control (Fig 2C). Interestingly, SM normalised the differential cell count of BAL fluid to the level of healthy animals, while Dex did not improve this parameter (Fig 2C). Evaluation of the pro-inflammatory cytokines in the lung tissue and BAL fluid showed that both anti-inflammatory compounds significantly reduced TNF-α and IL-1β expression in the ALI mice compared with the controls (Fig 2D).

Histological analysis revealed that the administration of anti-inflammatory drugs resulted in the abrogation of inflammatory changes in the lung tissue of LPS-challenged mice, while administration of vehicle did not affect these pathomorphological alterations. We observed a significant reduction in inflammatory infiltration, circulatory, and destructive changes; only slight interstitial oedema and single inflammatory cells with an absence of severe tissue damage can be seen in the lungs of LPS-induced animals treated with SM and Dex (Fig 2F, upper panel). PAS staining of neutral mucins in bronchial epithelium clearly demonstrated that SM and Dex effectively inhibited mucus hyperproduction, which reflects their anti-inflammatory action (Fig 2F, middle panel). Similarly, immunohistochemical staining of lungs with TNF-α antibodies shows that treatment of ALI mice with SM significantly decreased TNF-α expression in the lung tissue, almost to the level of healthy mice; this was not noted with Dex (Fig 2F, bottom panel).

These results clearly show that LPS effectively inflames lung tissue *in vivo* and this reaction is substantially reversed by anti-inflammatory compounds with different mechanisms of

action. Thus, the described murine model of ALI can be used to validate the role of key genes in the development of pulmonary inflammation and evaluate their susceptibility to anti-inflammatory therapy.

## Analysis of gene expression patterns associated with ALI development

The expression levels of *Il-6*, *Ccl2*, *Cat*, *Serpine1*, *Eln*, *Timp1*, *Ptx3*, and *Socs3* (Fig 1E, under-lined genes), the DEGs identified as master regulators of ALI using bioinformatics approaches, were analysed by qRT-PCR and TaqMan probes in the lung tissue of LPS-challenged mice untreated or pre-treated with Dex or SM. Data are summarised in Table 2 and S1 Fig.

The lung tissue of healthy animals was characterised by very low expression levels of studied genes, which were set as 1 (Table 2, S1 Fig), whereas LPS challenge was found to cause a mani-fold increase of their expression. The change of the expression of the genes in ALI mice com-pared to healthy ones was as follows in descending order: *Il-6* (~260 fold) > *Ccl2* (~250 fold) > *Timp1* (~120 fold) > *Socs3* (~35 fold) > *Serpine1* (~25 fold) > *Ptx3* (~4 fold) > *Eln* (~2 fold) (Table 2, S1 Fig). The only down-regulated gene upon ALI development among the ana-lysed genes was *Cat* (~2 fold reduction) (Table 2, S1 Fig).

Pre-exposure administration of SM or Dex, together with the significant suppression of the development of LPS-induced inflammation in the lungs, prevented to some extent the up-reg-ulation of ALI associated genes (Table 2, S1 Fig), whereas administration of vehicle exhibited no effects on LPS-induced up-regulation of analysed DEGs (Table 2, S1 Fig).

The expression of genes encoding for the pro-inflammatory cytokines *Il-6* and *Ccl2* was decreased after SM and Dex administration, most significantly opposed to other up-regulated DEGs in the lung tissue of LPS-challenged mice, but did not reach the level of healthy animals (Table 2, S1 Fig). In addition, SM tended to decrease the expression of these genes more signif-icantly compared to Dex, although these differences were not statistically significant.

The decrease in the expression of other DEGs in the lung tissue of LPS-induced mice after administration of anti-inflammatory drugs was less significant than that of *Il-6* and *Ccl2*; the effect of SM and Dex decreased in the order *Timp1* > *Serpine1* > *Socs3* > *Eln* (Table 2, S1 Fig). The influence of SM or Dex on the expression of these genes was slightly different, but within the experimental errors. SM caused a more pronounced decrease in the expression of serine

**Table 2. The expression levels of genes identified as master regulators of ALI by bioinformatics approaches.**

| Gene ID | Microarray Data Fold Change* | | | | Experimental Data Fold Change* | | | |
|---|---|---|---|---|---|---|---|---|
| | GSE58654 Hyperoxia | GSE80011 Influenza | GSE130936 LPS | GSE94522 Bleomycin | ALI** | Vehicle** | SM** | Dex** |
| *Il-6* | 11.5 | 92.8 | 13.1 | 11.3 | 258.8 | 252.8 | 137.9 | 167.3 |
| *Ccl2* | 6.8 | 30.6 | 3.0 | 7.4 | 244.9 | 310.6 | 164.6 | 179.6 |
| *Cat* | -2.1 | -2.1 | -2.6 | -2.2 | -1.9 | -2.6 | -2.1 | -2.1 |
| *Serpine1* | 8.9 | 8.4 | 7.1 | 4.4 | 25.2 | 36.4 | 18.5 | 20.8 |
| *Eln* | 3.1 | -3.5 | -2.6 | 7.5 | 2.1 | 1.4 | -1.2 | -1.5 |
| *Timp1* | 14.1 | 19.6 | 9.9 | 7.3 | 117.8 | 154.6 | 108.2 | 98.2 |
| *Ptx3* | 7.6 | 19.4 | 5.5 | 12.0 | 3.5 | 3.9 | 4.0 | 3.4 |
| *Socs3* | 3.0 | 3.9 | 7.0 | 2.9 | 35.0 | 37.7 | 23.1 | 24.4 |

*Expression level data in the experimental groups were normalised to the expression level in healthy mice. Three samples from each experimental group were analysed in triplicate. Data are shown for control groups ALI and vehicle (LPS challenged mice without treatment and after vehicle administration, respectively) and experimental groups SM and Dex (LPS chellenged mice after SM or Dex administration, respectively). *Il-6* –Interleukine 6, *Ccl2* –C-C Motif Chemokine Ligand 2, *Cat*–Catalase, *Serpine1* –Serine Proteinase Inhibitor. Clade E. Member 1, *Eln*–Elastin, *Timp1* –Tissue Inhibitor of Matrix Metalloproteinase 1, *Ptx3* –Pentraxin 3, *Socs3* –Suppressor of cytokine signaling 3.

proteinase inhibitor *Serpine1* and suppressor of cytokine signalling *Socs3* compared to Dex, while Dex more significantly down-regulated *Timp1* and *Eln*, encoding for a matrix metallo-proteinase inhibitor and elastic fibres, respectively, compared to SM. Expression of *Ptx3*, encoding for an inflammatory pattern-recognition molecule, did not change in the lung tissue of LPS-challenged mice after SM or Dex treatment compared with the controls (Table 2, S1 Fig). Finally, the expression levels of the antioxidant molecule *Cat* did not differ significantly from that of the ALI group after anti-inflammatory treatment, but was slightly increased after SM and Dex administration compared to the vehicle group (Table 2, S1 Fig).

Thus, our findings show that the expression of a number of identified ALI-associated hub genes (*Il-6*, *Ccl2*, *Cat*, *Serpine1*, *Eln*, *Timp1*, and *Socs3*) varied significantly during the progression and preventive therapy of LPS-induced lung inflammation, which indicates the close involvement of studied genes in the inflammatory response and the ability to be effectively suppressed by anti-inflammatory treatment.

## Identification and validation of key genes related to COVID-19 in the in vivo model of LPS-induced ALI

In the next step of the study, we questioned whether the revealed ALI-associated key genes are involved in pathologic processes in the lungs in other severe pulmonary diseases, including COVID-19 and lung cancer. In order to answer the first question, 58 common DEGs involved in the regulation of pulmonary inflammation in mice (Fig 1A) were overlapped with genes implicated in the progression of COVID-19 in the lungs of patients, recently published by Daamen et al. [82]. The performed analysis showed that 15 of 58 ALI-associated DEGs changed expression during SARS-CoV-2-induced destructive changes in lung tissue, including 2 down- and 13 up-regulated genes (Fig 3A). Further reconstruction of the gene association network of the COVID-19-associated regulome and its analysis demonstrated that a range of revealed genes were characterised by a high degree centrality within the network, such as *SAA1*, *RSAD2*, *IFI44*, *RTP4*, *MMP8*, and *CCL2* (21–44 interconnections) that show their probable key regulatory roles in lung injury induced by SARS-CoV-2 infection (Fig 3B). These data are consistent with the data from the text mining analysis presented above, which demonstrated the co-occurrence of *CCL2* and *MMP8* with COVID-19 in the scientific literature deposited in the MEDLINE database (Fig 1C).

Next, in order to reveal biological processes in COVID-19-injured lungs in which 15 overlapping genes between ALI and COVID-19 (Fig 3A) may be involved, their first neighbours were identified from the COVID-19-associated regulome, followed by the functional annotation of the reconstructed network (Fig 3C). The analysis showed that ALI-associated genes and their partners are involved in the regulation of three main processes: responses to viral and bacterial infection (e.g., coronavirus disease, influenza A, cellular response to virus, *Staphylococcus aureus* infection, etc.), immune responses (e.g., interferon signalling, Toll-like receptor signalling pathway, inflammatory response, humoral immune response, etc.), and G protein-coupled receptor (GPCR) signalling pathways (e.g., GPCR ligand binding, signalling by GPCR, etc.) (Fig 3D). Given the revealed enrichment in terms associated with virus-induced immune responses as well as the known involvement of GPCR signalling in the progression of numerous inflammatory disorders [83], our analysis clearly shows the probable regulatory role of a range of ALI-related DEGs in lung injury mediated by SARS-CoV-2 infection. Thus, these results demonstrate a translational bridge between acute lung inflammation induced by LPS challenge in mice and COVID-19 in humans.

Furthermore, the top 5 ALI-related genes that were the most associated with the COVID-19 regulome (*Saa1* (28), *Rsad2* (15), *Ifi44* (14), *Rtp4* (10), *Mmp8* (13); scores are in accordance

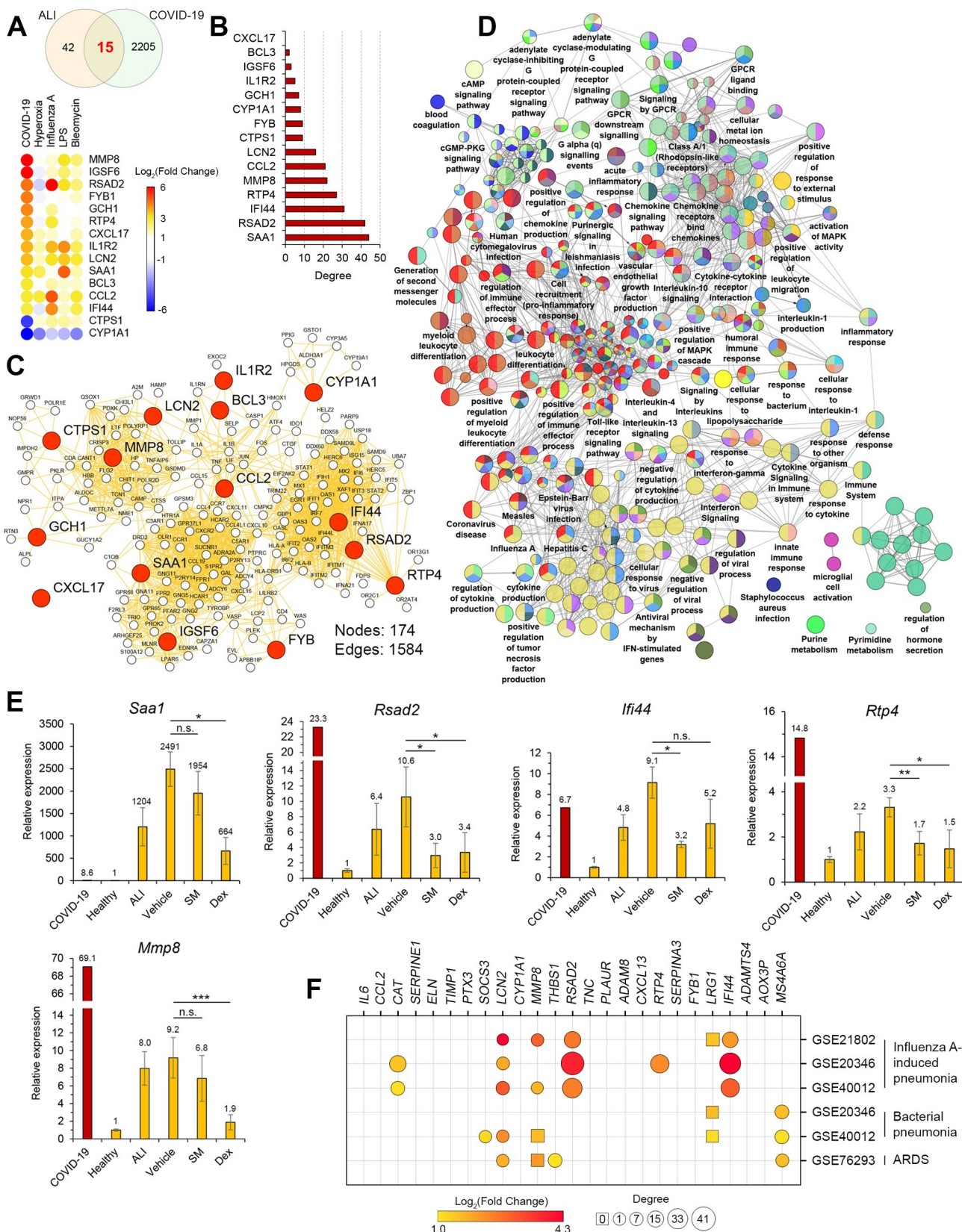

**Fig 3. Involvement of ALI-associated DEGs in the regulation of lung injury induced by SARS-CoV-2 infection in humans.** (**A**) Upper panel: Venn diagram of genes differentially expressed in the lungs injured by various irritants (ALI) or SARS-CoV-2 (COVID-19) in mice and humans, respectively. Lower panel: The heat map illustrating the expression of 15 overlapping genes between ALI and COVID-19 in analysed datasets. (**B**) The degree centrality values of ALI-associated DEGs in COVID-19-related gene association network. (**C**) The network of 15 overlapping genes between ALI and COVID-19 with their first neighbours within the COVID-19-related regulome. (**D**) Functional annotation of the network depicted in Fig 3C, using the ClueGO plugin in Cytoscape. (**E**) Expression of the top 5 ALI-associated DEGs, i.e. those most interconnected with the COVID-19-related regulome, in the lungs of healthy mice or mice with LPS-induced ALI without treatment and after SM or Dex administration. Relative expression levels of these genes were normalised to the expression level of hypoxanthine phosphoribosyltransferase (*HPRT*) (used as a reference gene). Three samples from each experimental group were analysed in triplicate. The data are shown as mean ± standard deviation. * $p<0.05$, ** $p<0.01$, *** $p<0.001$, n.s.—not significant. The red bar represents changes in the expression of the mentioned genes in the lungs of COVID-19 patients *vs.* healthy group, according to Daamen et al. [82]. (**F**) The expression levels and the degree centrality values of mice-specific ALI-related core genes in human ALI/ARDS induced by different stimuli.

with Fig 1D) were selected to analyse their expression in LPS-injured murine lungs and to assess whether anti-inflammatory therapy affects their expression. Treatment of mice with LPS significantly increased the expression of all these genes in lung tissues, which agrees well with their up-regulation observed in murine lungs injured by other triggers (Fig 3A). Interestingly, sesame oil used as a vehicle for dexamethasone and SM delivery was found to reinforce the LPS-stimulated expression of analysed genes, especially in the case of *Saa1* and *Ifi44* (Fig 3E). Pre-treatment of mice with both anti-inflammatory compounds markedly reduced the LPS-stimulated expression of ALI/COVID-19-related key genes, and Dex demonstrated more pronounced effects compared to SM, leading to statistically significant down-regulation of 4 of 5 analysed genes (*Saa1*, *Rsad2*, *Rtp4*, *Mmp8*), whereas administration of SM significantly decreased the expression of only 3 of 5 selected genes (*Rsad2*, *Ifi44*, *Rtp4*) (Fig 3E). Our data show that nodal genes activated during ALI/COVID-19 are susceptible to anti-inflammatory therapy, including Dex, successfully used in the clinic to treat COVID-19 [84], and SM, a structural analogue of which has been studied in Phase 2 clinical trials in COVID-19 patients [85]. These findings demonstrate the expediency of further considerations of ALI/COVID-19-related genes as promising molecular markers and potential therapeutic targets to reduce the severity of ARDS associated with SARS-CoV-2 infection.

Finally, we questioned whether revealed core genes involved in regulation of ALI in mice play an important role in SARS-CoV-2-independent ALI/ARDS in human. To understand this, six different whole genome expression profiles in blood samples of patients with ALI induced by influenza A and bacterial infections as well as ARDS were analyzed (S1 Table). As depicted in Fig 3F, a range of mice-specific ALI-associated core genes were up-regulated in ALI/ARDS in human and, moreover, displayed relatively high node degree centrality scores. Obtained results clearly demonstrate that: first, there is a certain translational bridge between regulomes of ALI in mice and human; second, the group of influenza A-induced ALI in patients is the most enriched with mice-specific ALI-related key genes compared to groups of bacterial pneumonia and ARDS (Fig 3F); and, third, innate immunity regulators *RSAD2*, *IFI44* and *RTP4*, showing key nodal positions within both influenza A- (Fig 3F) and SARS-CoV-2-related (Fig 3B) gene networks, can be considered as novel key regulators of virus-induced ALI/ARDS in human.

## Association of identified ALI-related key genes with tumour transformation in the lungs

For bronchial and lung neoplasms, three pre-invasive lesions are recognised: (1) bronchial dysplasia for lung squamous cell carcinoma (LUSC), (2) atypical adenomatous hyperplasia for lung adenocarcinoma (LUAD) and (3) diffuse idiopathic pulmonary neuroendocrine hyperplasia for carcinoids [86]. It is believed that LUSC develops through the sequence of squamous

metaplasia, dysplasia, and carcinoma *in situ* caused by direct exposure of the respiratory epithelium to damaging agents [87]. One of the long-term effects of SARS-CoV-2-induced lung inflammation has been found to be type 2 pneumocyte hyperplasia and squamous metaplasia with atypia in the small bronchi, which are premalignant lesions associated with a high risk of squamous cell carcinoma of the lung [88–90]. Moreover, similar changes in the lung tissue can develop as a result of inflammation and regeneration disorders caused by other damaging factors, such as cigarette smoke [91], influenza infection [89, 92], chemicals [93], vitamin A deficiency [94], and in general can be the outcome of pulmonary fibrosis of any origin [95, 96]. LUAD originates from type 2 pneumocytes or Clara cells and arises through a stepwise process from atypical adenomatous hyperplasia to adenocarcinoma *in situ*, minimally invasive adenocarcinoma, and eventually overt invasive adenocarcinoma; the risk factors are similar to those of LUSC [97–99].

Taking into account the close relationship between acute and chronic inflammation, pulmonary fibrosis, metaplasia, dysplasia, and malignant transformation, the associations of the identified ALI-related genes with the survival of patients with LUAD and LUSC were analysed using The Cancer Genome Atlas (TCGA) database (Figs 4 and S2). Our analysis shows that 14 of the top 25 key genes associated with acute lung inflammation (Fig 1) can affect lung cancer severity, and changes in their expression levels can be associated with a poor prognosis for LUAD and LUSC patients. It was found that up-regulation of *IL-6*, *CCL2*, *PTX3*, *TIMP1*, *SERPINE1*, *MMP8*, *PLAUR*, *ADAM8*, *THBS1*, and *SERPINA3* was associated with poor survival in patients with LUSC (Fig 4A and 4B, S2 Fig), whereas high expression of *CXCL13*, *PTX3*, *TIMP1*, *SERPINE1*, *PLAUR*, and *ADAMTS4* were correlated with high mortality rate of patients with LUAD (Figs 4A, 4B and S2). Overlapping these gene sets revealed that 4 genes (*PTX3*, *TIMP1*, *SERPINE1*, and *PLAUR*) are common to both LUSC and LUAD patients, which may indicate the universality of these markers for the occurrence of lung malignant neoplasms (Figs 4A, 4B and S2). Additionally, it was found that low expression of *CAT* and *ELN* is associated with poor LUAD survival (Figs 4A, 4B and S2).

Next, in order to verify the obtained genomic data, the protein expression of PTX3, TIMP1, SERPINE1, and PLAUR, whose mRNA expression was found to be correlated with high mortality of both LUSC and LUAD patients (TCGA data) (Fig 4B), was analysed using The Human Protein Atlas database (Fig 4C). Obtained results clearly demonstrate the importance of mentioned regulators for LUAD/LUSC progression: it was shown that up-regulation of PTX3, TIMP1, SERPINE1, and PLAUR not only on mRNA, but also on protein level was significantly associated with poor survival in patients with lung cancers (Fig 4C).

Moreover, it is clearly shown that most of the studied genes (*IL-6*, *CXC13*, *TIMP1*, *CCL2*, *ADAM8*, *THBS1*, *SOCS3*, *SERPINA3*, *SERPINE1*, *PLAUR*, *TNC*, *MMP8*, *IFI44*, *RSAD2*, *PTX3*, *LRG1*, *ELN*) are involved in pathogenesis and have prognostic value for a wide variety of non-tumour chronic lung pathologies which can serve as a background for malignant transformation, such as idiopathic pulmonary fibrosis [100, 101], chronic bronchitis, emphysema and chronic obstructive pulmonary disease (COPD) [102, 103], bronchiectasis and cystic fibrosis [104, 105], systemic sclerosis accompanied with the pulmonary fibrosis [106, 107] (all found associations of ALI-related core genes/proteins with nonmalignant chronic lung diseases were summarized in S2 Table).

Thus, our findings demonstrate that the identified ALI-associated key genes play an important role, not only in the acute lung inflammation caused by various etiological factors, but also have a close connection with chronic lung pathology and malignant transformation of lung tissue, perhaps due to the intercommunity of signalling pathways regulating these processes.

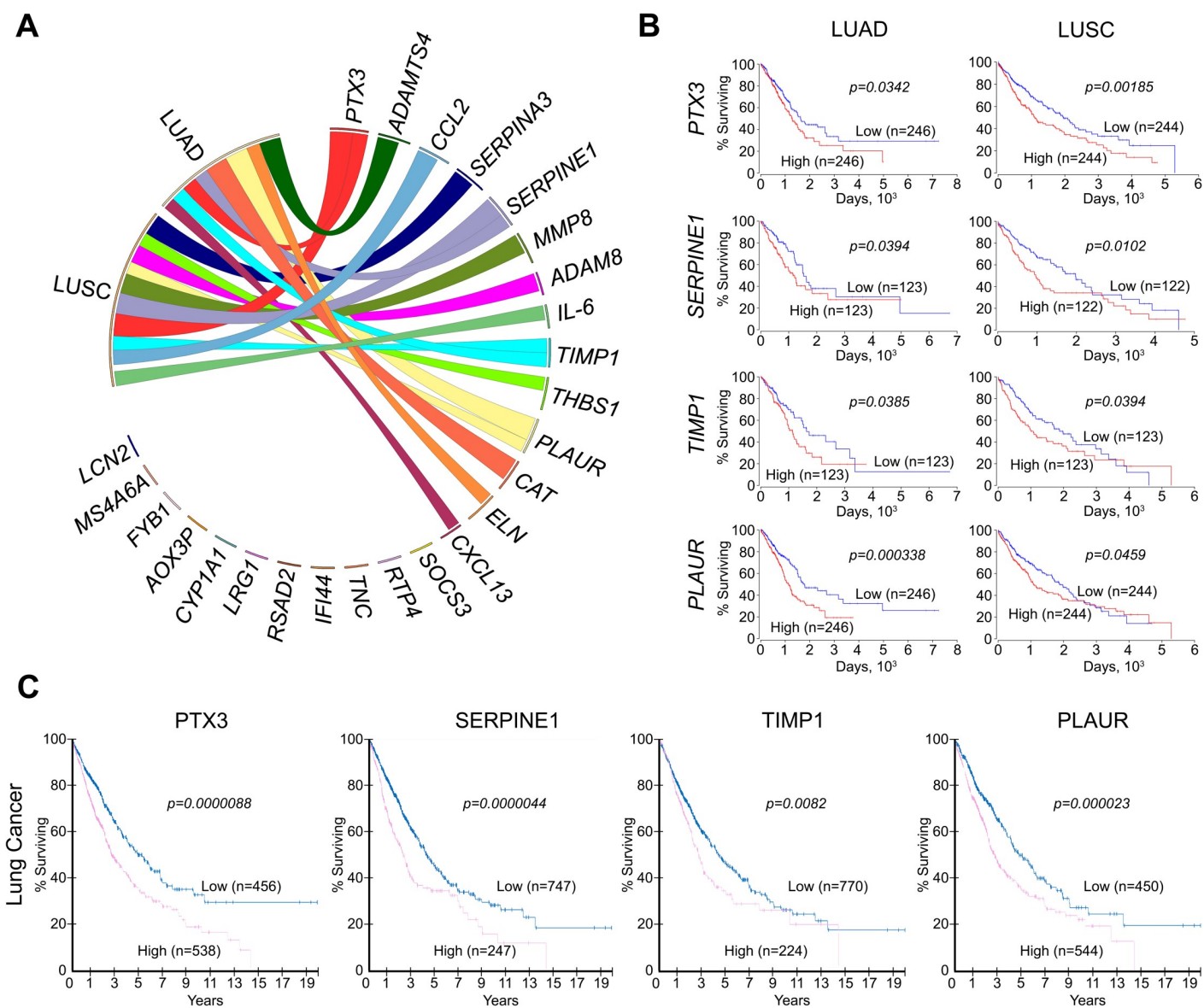

**Fig 4. Association of identified DEGs with survival of patients with lung adenocarcinoma (LUAD) and lung squamous cell carcinoma (LUSC).** (**A**) Survival analysis of DEGs was performed using The Cancer Genome Atlas (TCGA) data for patients with LUAD and LUSC. Visualisation was performed via Circos. The width of ribbons corresponds with the log-rank *p*-value, with wider ribbons indicating a more significant correlation. The ribbons of *ELN* and *CAT* indicate that low expression of these genes is associated with poor survival of patients, as opposed to the other genes, for which high expression is associated with poor survival. The absence of ribbons indicates no significant correlation between gene expression and patient survival. (**B**) Survival of patients with LUAD and LUSC depending on the level of particular gene expression in the lung tissue (mRNA level). Kaplan-Meier survival curves were constructed based on TCGA data using OncoLnc. (**C**) Survival of patients with lung cancers depending on the level of particular gene expression in the lung tissue (protein level). Kaplan-Meier survival curves were constructed based on The Human Protein Atlas data.

## Discussion

Acute lung inflammation is known to be one of the main pathogenetic components of ALI/ARDS and one of the manifestations of a large number of respiratory disorders, including COVID-19 [15, 108]. In the absence of appropriate therapy, this pathology often becomes chronic, leading irreversible scarring and remodelling of the lung tissue with fibrosis formation and/or malignant transformation [109, 110]. Regulation of such complex consequences of

the acute lung damage may restore unbalanced lung homeostasis, reducing the risk of lung function decline and irreversible changes in the lungs.

Within the framework of this idea, identification of key elements of the regulatory network involved in the development of lung injury both in the acute phase and in the long-term phase and associated with the diverse etiological factors was the main objective of our study. Although numerous studies have been carried out to explore the pathogenesis of ALI/ARDS, including those caused by SARS-CoV-2 infection, it has still not been elucidated completely. In this work, we performed a comprehensive bioinformatics analysis with subsequent validation of the revealed master regulators on the *in vivo* ALI model to find target genes and pathways involved in the development of lung injury and to elucidate their relationships with COVID-19 and lung cancers.

Focusing our attention on the prevalent molecular pathways in different clinical conditions, we analysed four independent cDNA microarray datasets concerning hyperoxia-, influenza-, LPS-, and bleomycin-induced ALI and revealed the set of core genes involved in the regulation of acute pulmonary inflammation and characterised by nodal dispositions within ALI-related gene networks. A total of 58 DEGs consisting of 48 up-regulated and 10 down-regulated genes were identified between damaged and normal lung tissues. The functional enrichment analyses demonstrated that the identified DEGs were enriched in some biological processes such as the regulation of cytokine production, type 2 immune response, cellular response to interleukin-1, and lung fibrosis.

The identified top 25 key genes that were the most interconnected within the ALI-related regulome were associated with two main processes, including extracellular matrix organisation (*Adam8*, *Adamts4*, *Eln*, *Il6*, *Mmp8*, *Ptx3*, *Serpine1*, *Thbs1*, *Timp1*, *Tnc* (GO:0030198, *p = 1.564E-11*) and defence responses (*Adam8*, *Adamts4*, *Ccl2*, *Cxcl13*, *Il6*, *Lcn2*, *Mmp8*, *Ptx3*, *Rsad2*, *Rtp4*, *Serpina3*, *Serpine1*, *Socs3*, *Thbs1*, *Timp1* (GO:0006952, *p = 1.151E-10*)). The results agreed well with published data:

a. lung remodelling and activation of the innate immune response are known to be integral processes involved in ALI progression [111, 112];

b. nine of the top 25 ALI-related key genes (*Adamts4*, *Ccl2*, *Cxcl13*, *Il6*, *Lcn2*, *Ptx3*, *Serpina3*, *Socs3*, *Thbs1*) were recently identified by Liang et al. as nodal DEGs in the gene regulatory network of LPS-induced ALI in mice [113]. Moreover, the bioinformatics data reported by Tu et al. clearly demonstrated the key regulatory role of six of the identified key genes (*Ccl2*, *Il6*, *Lcn2*, *Rsad2*, *Socs3*, *Timp1*) in a model of mechanical ventilation-driven ALI in mice [114];

c. the majority of the top 25 key ALI-associated genes (19 of 25 genes) are involved in the rodent inflammatome signature [54] (Fig 5A); moreover, their degree centrality was mostly similar in the structures of both the ALI-related and rodent inflammatory gene networks (Fig 5A), which clearly shows a key contribution of inflammation to ALI progression.

Based on the data from the text mining analysis, eight genes (*Il-6*, *Timp1*, *Ccl2*, *Socs3*, *Serpine1*, *Ptx3*, *Cat*, and *Eln*) found to be most associated with ALI-related terms in the scientific literature were verified in a murine ALI model and found to respond well to LPS challenge and therapy with anti-inflammatory compounds. Also, an association of these genes with tumour transformation in the lungs both at the transcript and protein levels was found, which indicates their involvement in the delayed consequences of lung damage.

Further overlapping of ALI-associated nodal genes with gene expression profiling data from ARDS induced by COVID-19 in patients [82] revealed a range of common genes

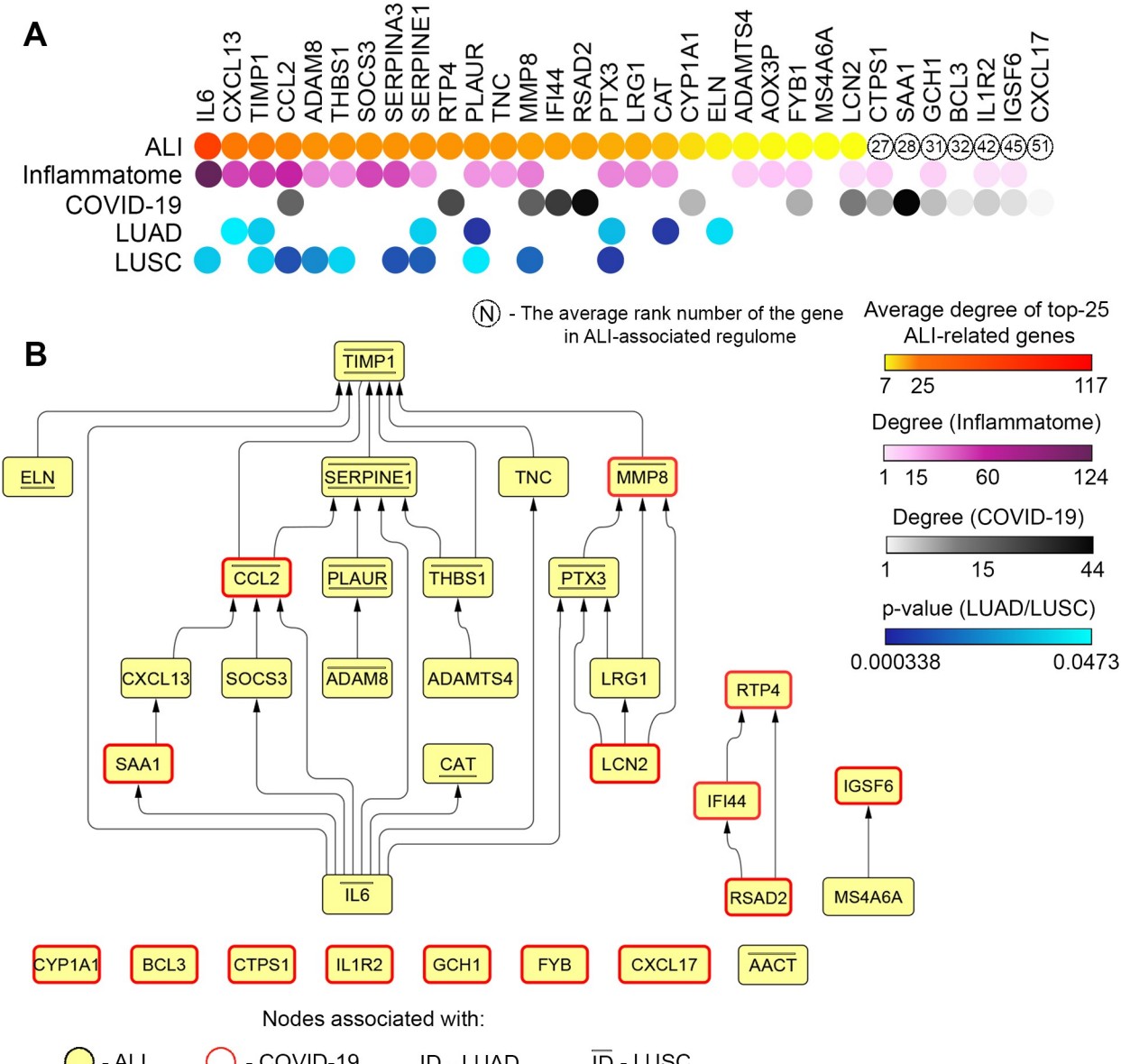

**Fig 5. Interconnections and interplay of revealed key ALI-associated genes in different pulmonary disorders.** (A) The heatmap showing interconnections of revealed ALI-associated key genes with the rodent inflammatome, the COVID-19 regulome, and the progression of lung cancer in patients. Degree: the number of linkages of the gene with partner genes within the gene association network. In the case of LUAD/LUSC, only those genes are marked in which the expression level is significantly and negatively associated with overall survival time for patients with lung cancer. (B) The hierarchical network reconstructed with all revealed key regulatory genes involved in ALI development (STRING, confidence score ≥ 0.7).

differentially expressed in both the acute (ALI, mice) and the late (COVID-19, postmortem, human) phases of lung injury (*CCL2, RTP4, MMP8, IFI44, RSAD2, CYP1A1, FYB1, LCN2*), which demonstrates their probable key regulatory role in the progression of ALI (Fig 5A). The RT-PCR analysis showed that a number genes that were highly interconnected with the COVID-19-related gene network (*Saa1, Rsad2, Ifi44, Rtp4,* and *Mmp8*) were up-regulated in lung tissue of mice in response to LPS and significantly down-regulated by pre-treatment of LPS-challenged mice with Dex and SM, two anti-inflammatory compounds, displaying promising efficiency against ARDS induced by SARS-CoV-2 [84, 85]. These data suggest that this

murine model of LPS-induced ALI is consistent in some respects with COVID-19-associated lung injury and may provide an effective tool for the development of an anti-inflammatory approach to COVID-19 therapy.

Interestingly, three ALI-associated key genes that were the most interconnected with the COVID-19-related regulome (*RTP4*, *IFI44*, *RSAD2*) were not involved in the inflammatome signature and had no associations with tumour transformation in the lungs (Fig 5A). It is known that *RTP4*, *IFI44*, and *RSAD2* are interferon-inducible genes that participate in the regulation of the host antiviral response [115–117]. Our results show that these genes, on the one hand, form a common cluster with partner genes within the COVID-19-related gene regulatory network (Fig 3C), and, on the other hand, are not associated with ALI/ARDS, according to our text mining analysis (Fig 1E). Given these findings as well as the reported involvement of *RTP4*, *IFI44*, and *RSAD2* in regulation of the host response to SARS-CoV-2 infection [118, 119], these genes may be considered as probable novel markers of pulmonary inflammatory disorders. Two other genes, *CCL2* and *MMP8*, displayed a high degree centrality scores in both ALI- and COVID-19-related gene networks as well as the inflammatome (Fig 5A), and are known important regulators of neutrophil-driven repair of injured lungs, as these protein products have neutrophil chemoattractant function [120] and are involved in the remodelling of damaged lung tissues [26], respectively. The observed change in the degree centrality score of *SAA1* from 5 to 44 in ALI- and COVID-19-associated regulomes, respectively, demonstrates the expediency of further investigations into *SAA1* as a probable marker of the severity of lung injury (Fig 5A). SAA1 is known to be a major constituent of the acute-phase proteins, which is chemotactic for several subtypes of leukocytes and promotes monocyte survival [121]. Moreover, it was found that recombinant SAA1 is a potent inducer of G-CSF *in vivo*, which leads to neutrophilia [122], the immoderate activation of which is tightly associated with ARDS [123].

Despite the fact that expression of revealed ALI/COVID-19-related core genes has been analysed in post-mortem lung tissues of COVID-19 patients [82], it was recently shown that a range of these genes (*CCL2*, *MMP8*, *IFI44*, *LCN2*, *SAA1*) were also differentially expressed in plasma, BALF and nasopharyngeal/saliva samples of COVID-19 patients and were significantly correlated with the severity of COVID-19 and unfavorable clinical outcomes [118, 124–127]. *RSAD2* and *IL1R2* were also found to be involved in regulation of SARS-CoV-2 infection, however their expression were positively correlated only with viral load [128] and viral shedding time [129], whereas their association with COVID-19 severity has not yet been shown. Interestingly, the listed genes reported as potent markers of critical illness in COVID-19 (*CCL2*, *MMP8*, *IFI44*, *LCN2*, *SAA1*) were identified by us as key regulators of both murine ALI and COVID-19-associated ARDS in human (Fig 5A) with the highest scores of degree centrality. Thus, obtained results clearly confirm the fact that gene network analysis data can be used as a valuable source to find novel discriminators of disease progression.

Our findings also demonstrate that the majority of the revealed key genes involved in the regulation of ALI are correlated with a poor prognosis for patients with LUAD and LUSC (Fig 5A). Interestingly, four genes (*TIMP1*, *SERPINE1*, *PLAUR* and *PTX3*) associated with poor overall survival of both LUAD and LUSC patients, are involved in the regulation of ECM remodelling [130–133], which plays a key role in the aggravation of tumour malignancy and metastasis [134]. Given the well-known induction of ALI in cancer patients in response to chemotherapy (e.g. gefitinib [135], gemcitabine [136], 5-fluorouracil [137]) or lobectomy for non-small cell lung cancer [138], our findings show that ALI-associated nodal genes can be used as molecular markers to control the severity of probable lung cancer recurrence after surgical treatment and chemotherapy.

As a result of the comprehensive bioinformatics analysis performed in this study, we identified a range of master regulators involved in the development of ALI in mice and displaying

translational implications in some severe pulmonary disorders in humans. Due to the high diversity of revealed key genes, we further questioned which ones are the most crucial for the regulation of ALI progression. To understand this, a hierarchical gene association network was reconstructed from the identified master regulatory genes (Fig 5B). As shown in Fig 5B, *IL6* is located at the most upstream position in the gene network, which clearly demonstrates the key functions of this interleukin in ALI development and is completely consistent with the highest degree centrality of *IL6* in the analysed ALI-related regulomes (Fig 1D) and our text mining data, where IL6 was the most interconnected gene/protein with ALI/ARDS (Fig 1E). *TIMP1* was found to occupy the bottom position within the master regulatory gene network (Fig 5B). Given its hub role in the ALI-related PPI network (third degree centrality rank after *IL6* and *CXCL13*) (Fig 1D), our analysis showed the expediency of further study of *TIMP1* as a promising therapeutic target for ALI treatment as a nodal element of the ALI-associated gene network. Other genes that can be also considered as potential ALI-related targets are *SER-PINE1*, *CCL2*, *MMP8*, and *PTX3*, which, according to the reconstructed network, are linked with several ALI-associated master regulators (Fig 5B) and show relatively high degree centrality in the ALI-specific PPI networks (rank scores 9, 4, 13, and 16, respectively) (Fig 1D). Interestingly, the network depicted in Fig 5B can be divided into two main regions: upstream positions within the network are occupied by genes (*IL6*, *SAA1*, *LCN2*, *SOCS3*, *CXCL13*) involved in the regulation of inflammation, whereas the majority of downstream genes are known regulators of ECM remodelling (*TIMP1*, *ELN*, *SERPINE1*, *TNC*, *MMP8*, *CCL2*, *PLAUR*, *THBS1*, *PTX3*, *ADAM8*, *ADAMTS4*). The results clearly show that the inflammatory processes is a key trigger of ALI development and its pharmacological suppression can be an effective approach to controlling the first steps of lung injury, in agreement with published data [139]. In the case of already developed ALI, the inhibition of ECM remodelling, according to our findings, may be a promising therapeutic strategy to ameliorate this pathology. Indeed, previous works have shown that the inhibition of matrix metalloproteinases, which play a key role in ECM modulation, effectively inhibited ALI in different *in vivo* models [140–142]. Interestingly, the majority of revealed ALI-associated DEGs involved in the regulation of SARS-CoV-2-induced ARDS either formed distinct modules in the master regulatory network (*RSAD2*, *IFI44*, *RTP4*) or did not have links with the network at all (Fig 5B). The low enrichment of the master regulatory network with COVID-19-associated key genes (Fig 5B) may demonstrate that ALI, despite being a useful experimental tool to develop novel therapeutics for the treatment of COVID-19, should however be used with certain limitations since it does not fully model the late changes in lung tissues observed in the postmortem lungs of COVID-19 patients.

## Conclusions

The integrative bioinformatics analysis of four independent ALI-associated cDNA microarray datasets with subsequent verification using RT-PCR data revealed a set of core genes involved in the regulation of acute pulmonary inflammation in mice, which also play a key regulatory role in ARDS induced by COVID-19 and LUAD/LUSC progression in humans. Using a murine ALI model pre-treated with anti-inflammatory compounds (dexamethasone, soloxo-lone methyl), we revealed that the expression level of these genes is significantly correlated with the severity of ALI and, therefore, these ALI-associated key genes and their protein products should be further investigated as promising therapeutic targets to control pulmonary inflammatory disorders. Our findings clearly show that *Rtp4*, *Ifi44*, *Rsad2*, and *Saa1* can be considered novel marker genes of ALI. Network analysis demonstrated that inflammation and subsequent ECM remodelling in lung tissue are key steps of ALI development, and

pharmacologically targeting key nodes involved in these processes (inflammation: *Il6*, ECM modulation: *Timp1*, *Serpine1*, *Ccl2*, *Mmp8*, *Ptx3*) may be a promising therapeutic strategy to treat ALI/ARDS. The involvement of some ALI-associated core genes in COVID-19-related regulome shows that the ALI model can be used as a promising tool to develop anti-inflammatory approaches targeting the early steps of lung injury induced by SARS-CoV-2. Finally, considering the identified association of ALI-specific key genes with poor prognosis in LUAD/LUSC patients, *TIMP1*, *SERPINE1*, *PLAUR*, and *PTX3* may be used as marker genes to control the severity of lung cancer progression in the case of accompanying pulmonary inflammation.

## Supporting information

**S1 Fig. Expression levels of key genes identified by bioinformatics analysis.** Data of qRT-PCR and TaqMan probes in the lung tissue of LPS-challenged mice without treatment and after SM and Dex administration. Relative expression levels were normalised to the level of hypoxanthine phosphoribosyltransferase (HPRT) (used as the reference gene). Three samples from each experimental group were analysed in triplicate. The data are shown as mean ± standard deviation. The statistical analysis was performed using the two-tailed unpaired t-test; $p$-values $< 0.05$ were considered as statistically significant.
(TIF)

**S2 Fig. Kaplan-Meier survival curves for patients with lung adenocarcinoma (LUAD) and lung squamous cell carcinoma (LUSC) constructed on the basis of The Cancer Genome Atlas (TCGA) data using the OncoLnc tool.**
(TIF)

**S1 Table. Datasets used in the bioinformatics analysis.**
(DOCX)

**S2 Table. Association of ALI-related core genes with chronic lung pathology in human.**
(DOCX)

## Acknowledgments

The authors gratefully thank Dr Oksana V. Salomatina and Prof Nariman F. Salakhutdinov (N.N. Vorozhtsov Novosibirsk Institute of Organic Chemistry SB RAS, Novosibirsk, Russia) for the provision of soloxolone methyl and Dr Elena L. Chernolovskaya (Institute of Chemical Biology and Fundamental Medicine SB RAS, Novosibirsk, Russia) for helpful discussion.

## Author Contributions

**Conceptualization:** Andrey V. Markov.

**Data curation:** Aleksandra V. Sen'kova, Evgenyi V. Brenner, Andrey V. Markov.

**Formal analysis:** Innokenty A. Savin, Evgenyi V. Brenner.

**Funding acquisition:** Marina A. Zenkova.

**Investigation:** Aleksandra V. Sen'kova, Innokenty A. Savin.

**Methodology:** Aleksandra V. Sen'kova, Andrey V. Markov.

**Project administration:** Marina A. Zenkova.

**Resources:** Marina A. Zenkova.

**Supervision:** Marina A. Zenkova, Andrey V. Markov.

**Validation:** Innokenty A. Savin.

**Visualization:** Aleksandra V. Sen'kova.

**Writing – original draft:** Aleksandra V. Sen'kova, Andrey V. Markov.

**Writing – review & editing:** Marina A. Zenkova.

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
