## [Decision Letter · Decision Letter 0]

12 Oct 2021

PONE-D-21-28077Core genes involved in the regulation of acute lung injury and their association with COVID-19 and tumor progression: a bioinformatics and experimental studyPLOS ONE

Dear Dr. Sen'kova,

Thank you for submitting your manuscript to PLOS ONE. After careful consideration, we feel that it has merit but does not fully meet PLOS ONE’s publication criteria as it currently stands. Therefore, we invite you to submit a revised version of the manuscript that addresses the points raised during the review process. Please submit your revised manuscript by Nov 26, 2021. If you will need more time than this to complete your revisions, please reply to this message or contact the journal office at plosone@plos.org. Please include the following items when submitting your revised manuscript:A rebuttal letter that responds to each point raised by the academic editor and reviewer(s). You should upload this letter as a separate file labeled 'Response to Reviewers'.A marked-up copy of your manuscript that highlights changes made to the original version. You should upload this as a separate file labeled 'Revised Manuscript with Track Changes'.An unmarked version of your revised paper without tracked changes. You should upload this as a separate file labeled 'Manuscript'.

We look forward to receiving your revised manuscript.

Kind regards,

You-Yang Zhao

Academic Editor

PLOS ONE

Journal Requirements:

2. We note that Figure 4c in your submission contain copyrighted images. All PLOS content is published under the Creative Commons Attribution License (CC BY 4.0), which means that the manuscript, images, and Supporting Information files will be freely available online, and any third party is permitted to access, download, copy, distribute, and use these materials in any way, even commercially, with proper attribution. For more information, see our copyright guidelines: http://journals.plos.org/plosone/s/licenses-and-copyright.

a) You may seek permission from the original copyright holder of Figure 4c to publish the content specifically under the CC BY 4.0 license. 

Reviewers' comments:

Reviewer's Responses to Questions

**Comments to the Author**

1. Is the manuscript technically sound, and do the data support the conclusions?

Reviewer #1: Partly

Reviewer #2: Yes

2. Has the statistical analysis been performed appropriately and rigorously? 

Reviewer #1: Yes

Reviewer #2: Yes

3. Have the authors made all data underlying the findings in their manuscript fully available?

Reviewer #1: Yes

Reviewer #2: Yes

4. Is the manuscript presented in an intelligible fashion and written in standard English?

Reviewer #1: Yes

Reviewer #2: Yes

5. Review Comments to the Author

Reviewer #1: This is an interesting story. The author used microarray datasets from GEO to identify common DEGs assosicted with ALI induced by different causes in mice models. Then the author validated their findings by establishing LPS-induced ALI mice model, while, treatment like Dex could downregulate the expression of the hub genes that were significantly increased in ALI. But, there are still some questions:

1.Can the mice model mimic human ALI? The common hub genes identified from GEO database were based on mice studies. Are those genes also the core genes for human ALI?

2.Some of the hub genes were also significant in COVID19 patients mentioned by other study. Did all the COVID19 patients have ALI? Whether the hub genes were correlated with disease severity? And how to validate that those hub genes could be the potential biomarker in COVID19-induced ALI. Can the hub genes be used to predict COVID19 patients with ALI from patients without ALI?

3.Those genes were associated with ALI, COVID19 and lung cancer, how about their role in other disease like COPD, pulmonary fibrosis? Is it possible that those genes are not only associated with ALI but some chronic injury like cancer.

Reviewer #2: This interesting manuscript entitled “Core genes involved in the regulation of acute lung injury and their association with COVID-19 and tumor progression: a bioinformatics and experimental study” is a systems biology study report to identify the core genes of ALI and the relationship to COVID-19. It uses transcriptomic analysis of existing datasets, text-mining of literature, animal models of acute lung injury, and other systems biology tools. It is novel study with some interesting conclusion. There is only some minor concerns:

1. One gene or one pathway can be dysregulated in multiple diseases, for example, the pathway inflammation can be found in all inflammation related disease. When the major conclusion is made, such as “The overlap with DEGs identified in postmortem lung tissues from COVID-19 patients revealed genes ( Saa1 , Rsad2 , Ifi44 , Rtp4 , Mmp8 ) that (a) showed a high degree centrality in the COVID-19-related regulatory network…”, can the “high degree” or statistical significance be defined?

2. The authors need to generate a table of used datasets with listed essential information, including how many subjects, what cohort, what tissue, et al. Why these datasets are picked up for the study, are they chosen because other datasets do not generate the same data? Please provide prioritization criteria.

6. PLOS authors have the option to publish the peer review history of their article (what does this mean?). If published, this will include your full peer review and any attached files.

Reviewer #1: **Yes: **Chao Du

Reviewer #2: No

---

## [Author Response · Author response to Decision Letter 0]

1 Nov 2021

Reviewers' comments:

Reviewer #1: This is an interesting story. The author used microarray datasets from GEO to identify common DEGs assosicted with ALI induced by different causes in mice models. Then the author validated their findings by establishing LPS-induced ALI mice model, while, treatment like Dex could downregulate the expression of the hub genes that were significantly increased in ALI. But, there are still some questions:

1. Can the mice model mimic human ALI? The common hub genes identified from GEO database were based on mice studies. Are those genes also the core genes for human ALI?

Dear Reviewer #1,

We are very grateful to you for the valuable comments and suggestions that helped us to improve the manuscript. We revised the manuscript according to your comments and, please, let us respond to your questions.

The mice model mimics human ALI well.

ALI/ARDS in human can be induced by various stimuli and accompany a huge number of human disease, like bacterial and viral pneumonia (bacterial, induced by Streptococcus pneumonia or Staphylococcus aureus [1,2]; viral, induced by influenza virus or rhinovirus [3,4]), ventilator-induced lung injury (VILI) [5–7], chemicals exposure such as chlorine, phosgene, and industrial pollutants [8–10], e-cigarette/vaping associated lung injury (EVALI) [11,12], traumatic brain injury-induced ALI [13,14], sepsis [15,16], severe acute pancreatitis-related ALI [17], etc. In in vivo studies, investigators try to simulate such disorders in experimental animals to induce ALI. A huge number of published works clearly demonstrated that rodent ALI reproduces well human ALI on different levels, including cytokine profile, cell populations in BALF and histological changes in lung tissue.

In order to develop effective therapeutic agents against ALI/ARDS, novel drug candidates are usually tested in rodent models of ALI during their preclinical studies. For instance, high anti-ALI efficiency of Iloprost, Sivelestat and Pirfenidone, being now at Phase 3-4 of clinical trials in patients with ARDS [18–20], have been firstly proven on various models of murine ALI before their clinical trial evaluation [21–23]. This fact can demonstrate not only a certain similarities between rodent model of ALI and ALI in patients, but also successful usage of these similarities in drug discovery and effective ALI/ARDS treatment.

In order to analyze whether murine model of ALI mimics human ALI in gene network level, six different cDNA microarray datasets of ALI induced in human by influenza A (GSE21802, GSE20346, GSE40012) or bacterial (GSE20346, GSE40012) infection and human ARDS (GSE76293) were analysed and the expression level and node degree centrality scores of revealed mouse-specific ALI-associated core genes in human ALI regulomes were computed. Our findings showed the presence of certain translational bridge between murine and human ALI regulomes – a range of mouse-specific core genes were found to be up-regulated in human ALI and, moreover, were characterized by high degree centrality (such as RSAD2, IFI44, RTP4) (please, see Fig 3F). According to our analysis, not all identified ALI-associated key genes were revealed in human ALI transcriptome (only 10 of 25 genes) (Fig 3F). We suppose that such relatively low enrichment with the key genes in human ALI can be explained by different sources of biomaterial analysed by cDNA microarray assays – lung tissue in the case of murine ALI and blood in the case of human ALI. Anyway, our bioinformatics analysis demonstrates that mice model of ALI mimics human ALI, at least in part.

These data were included in Fig 3 and in the text of manuscript (please, see Fig 3F and lines 123-131 (Materials and Methods section) and lines 590-601 (Results section)).

2. Some of the hub genes were also significant in COVID19 patients mentioned by other study. Did all the COVID19 patients have ALI? Whether the hub genes were correlated with disease severity? And how to validate that those hub genes could be the potential biomarker in COVID19-induced ALI. Can the hub genes be used to predict COVID19 patients with ALI from patients without ALI?

Thank you for your valuable comment! Indeed, it is our omission to identify ALI-associated COVID-19-related core genes but not to discuss the correlation of their expression with the severity of COVID-19.

SARS-CoV-2 primarily targets the lung tissue by causing diffuse alveolar damage and may result in acute respiratory distress syndrome (ARDS) [24–26]. ARDS is a severe complication of COVID-19 pneumonia which develops during the advanced stage of COVID-19, with a mortality rate amounting to 34–50% in moderate and severe ARDS [27,28]. Of those patients hospitalized with COVID-19, approximately one-third develop ARDS [29].

Deep analysis of published data showed that a range of revealed core genes associated with both ALI and COVID-19 (CCL2, MMP8, IFI44, LCN2, SAA1) are significantly correlated with unfavorable clinical outcomes in patients with COVID-19 and can be used as markers of severity of this disease (i.e. ARDS) [30–34]. Residual ALI-related core genes were correlated with SARS-CoV-2 replication only and did not associated with survival of COVID-19 patients. This information was added to Discussion section (please, see lines 764-776).

In order to validate hub genes as probable biomarkers in COVID-induced ALI, their expression should be evaluated in blood/salivary/BALF samples of patients with clinically characterized different stages of COVID-19 followed by calculation of Pearson or Spearman correlation between gene expression levels and the severity of COVID-19. We hope that our findings will be helpful for diagnosticians to develop novel diagnostic assays to identification of severe COVID-19 patients at early stages in order to correct their therapy.

3. Those genes were associated with ALI, COVID19 and lung cancer, how about their role in other disease like COPD, pulmonary fibrosis? Is it possible that those genes are not only associated with ALI but some chronic injury like cancer.

It is clearly shown that most of the studied genes (IL-6, CXC13, TIMP1, CCL2, ADAM8, THBS1, SOCS3, SERPINA3, SERPINE1, PLAUR, TNC, MMP8, IFI44, RSAD2, PTX3, LRG1, ELN) are involved in pathogenesis and have prognostic value for a wide variety of non-tumour chronic lung pathologies which also can serve as a background for malignant transformation in the lungs, such as idiopathic pulmonary fibrosis, chronic bronchitis and chronic obstructive pulmonary disease (COPD), bronchiectasis and cystic fibrosis, emphysema, and systemic sclerosis accompanied with the pulmonary fibrosis. In addition, a number of studies have shown that targeting these genes involved in the progression of chronic lung pathologies, e.g. pulmonary fibrosis and COPD, is essential for the treatment of these diseases.

We included data on the relationship between the studied genes and chronic lung pathologies in the text of the manuscript and in the S2 Table (please, see lines 657-665, line 668 and S2 Table).

Reviewer #2: This interesting manuscript entitled “Core genes involved in the regulation of acute lung injury and their association with COVID-19 and tumor progression: a bioinformatics and experimental study” is a systems biology study report to identify the core genes of ALI and the relationship to COVID-19. It uses transcriptomic analysis of existing datasets, text-mining of literature, animal models of acute lung injury, and other systems biology tools. It is novel study with some interesting conclusion. There is only some minor concerns:

1. One gene or one pathway can be dysregulated in multiple diseases, for example, the pathway inflammation can be found in all inflammation related disease. When the major conclusion is made, such as “The overlap with DEGs identified in postmortem lung tissues from COVID-19 patients revealed genes ( Saa1 , Rsad2 , Ifi44 , Rtp4 , Mmp8 ) that (a) showed a high degree centrality in the COVID-19-related regulatory network…”, can the “high degree” or statistical significance be defined?

Dear Reviewer #2,

Thank you for the valuable comments and suggestions that helped us to improve the manuscript. In our network analysis, the cut-off criterion to identify genes as key/core genes is its degree more than 10 within the analyzed network. Thus, the phrase “gene with a high degree centrality” means that this core gene has at minimum 10 gene neighbors in the regulome. This criterion is widely used in various published works in the field of gene network analysis (please, see some references [35–37]).

The phrase describing the criterion of core genes identification was added in the Section PPI Network Reconstruction in Materials and Methods (please, see lines 155-156).

2. The authors need to generate a table of used datasets with listed essential information, including how many subjects, what cohort, what tissue, et al. Why these datasets are picked up for the study, are they chosen because other datasets do not generate the same data? Please provide prioritization criteria.

We generate a table of used datasets with listed essential information (platform, induction stimuli, time after induction, number of samples, cohort, and tissue) (please, see S1 Table).

All of the analysed datasets on mice pathology concern processes associated with acute lung injury in mice at a time point earlier than 24 h after the induction, but caused by different stimuli (hyperoxia, influenza, LPS, bleomycin). Datasets on human pathology concern processes associated with acute lung injury in humans at day 1 after admission and also caused by different aetiological factors (influenza A-induced pneumonia, bacterial pneumonia, ARDS). So, datasets containing experimental and clinical data with ALI inducers of diverse origin, but on the early stage of the disease were analysed simultaneously in order to identify core genes common for acute lung pathology of various aetiologies.

References

1. Gonzales J, Chakraborty T, Romero M, Mraheil MA, Kutlar A, Pace B, et al. Streptococcus pneumoniae and Its Virulence Factors H2O2 and Pneumolysin Are Potent Mediators of the Acute Chest Syndrome in Sickle Cell Disease. Toxins. NLM (Medline); 2021. doi:10.3390/toxins13020157

2. Lucas R, Hadizamani Y, Gonzales J, Gorshkov B, Bodmer T, Berthiaume Y, et al. Impact of bacterial toxins in the lungs. Toxins (Basel). 2020;12. doi:10.3390/toxins12040223

3. Klomp M, Ghosh S, Mohammad S, Nadeem Khan M. From virus to inflammation, how influenza promotes lung damage. Journal of Leukocyte Biology. John Wiley and Sons Inc.; 2020. doi:10.1002/JLB.4RU0820-232R

4. Shah RD, Wunderink RG. Viral Pneumonia and Acute Respiratory Distress Syndrome. Clin Chest Med. 2017;38: 113. doi:10.1016/J.CCM.2016.11.013

5. Zhang H, Dong W, Li S, Zhang Y, Lv Z, Yang L, et al. Salidroside protects against ventilation-induced lung injury by inhibiting the expression of matrix metalloproteinase-9. Pharm Biol. 2021;59: 760. doi:10.1080/13880209.2021.1967409

6. Goligher EC, Douflé G, Fan E. Update in Mechanical Ventilation, Sedation, and Outcomes 2014. https://doi.org/101164/rccm201502-0346UP. 2015;191: 1367–1373. doi:10.1164/RCCM.201502-0346UP

7. Agrawal DK, Smith BJ, Sottile PD, Albers DJ. A Damaged-Informed Lung Ventilator Model for Ventilator Waveforms. Front Physiol. 2021;12. doi:10.3389/fphys.2021.724046

8. Radbel J, Laskin DL, Laskin JD, Kipen HM. Disease-modifying treatment of chemical threat agent–induced acute lung injury. Annals of the New York Academy of Sciences. Blackwell Publishing Inc.; 2020. pp. 14–29. doi:10.1111/nyas.14438

9. Pauluhn J. Phosgene inhalation toxicity: Update on mechanisms and mechanism-based treatment strategies. Toxicology. Elsevier Ireland Ltd; 2021. doi:10.1016/j.tox.2021.152682

10. Laskin DL, Malaviya R, Laskin JD. Role of Macrophages in Acute Lung Injury and Chronic Fibrosis Induced by Pulmonary Toxicants. Toxicological Sciences. Oxford University Press; 2019. pp. 287–301. doi:10.1093/toxsci/kfy309

11. Triantafyllou GA, Tiberio PJ, Zou RH, Lamberty PE, Lynch MJ, Kreit JW, et al. Vaping-associated Acute Lung Injury: A Case Series. https://doi.org/101164/rccm201909-1809LE. 2019;200: 1430–1431. doi:10.1164/RCCM.201909-1809LE

12. Kasson E, Singh AK, Huang M, Wu D, Cavazos-Rehg P. Using a mixed methods approach to identify public perception of vaping risks and overall health outcomes on Twitter during the 2019 EVALI outbreak. Int J Med Inform. 2021;155: 104574. doi:10.1016/j.ijmedinf.2021.104574

13. Zhang C-N, Li F-J, Zhao Z-L, Zhang J-N. The role of Extracellular Vesicles in Traumatic Brain Injury-induced Acute Lung Injury. https://doi.org/101152/ajplung000232021. 2021 [cited 25 Oct 2021]. doi:10.1152/AJPLUNG.00023.2021

14. Ziaka M, Exadaktylos A. Brain–lung interactions and mechanical ventilation in patients with isolated brain injury. Crit Care. 2021;25: 358. doi:10.1186/S13054-021-03778-0

15. Sever IH, Ozkul B, Erisik Tanriover D, Ozkul O, Elgormus CS, Gur SG, et al. Protective effect of oxytocin through its anti-inflammatory and antioxidant role in a model of sepsis-induced acute lung injury: Demonstrated by CT and histological findings. Exp Lung Res. 2021; 1–10. doi:10.1080/01902148.2021.1992808

16. Jiao Y, Zhang T, Zhang C, Ji H, Tong X, Xia R, et al. Exosomal miR-30d-5p of neutrophils induces M1 macrophage polarization and primes macrophage pyroptosis in sepsis-related acute lung injury. Crit Care. 2021;25: 356. doi:10.1186/S13054-021-03775-3

17. Kong L, Deng J, Zhou X, Cai B, Zhang B, Chen X, et al. Sitagliptin activates the p62–Keap1–Nrf2 signalling pathway to alleviate oxidative stress and excessive autophagy in severe acute pancreatitis-related acute lung injury. Cell Death Dis. 2021;12: 928. doi:10.1038/s41419-021-04227-0

18. Iloprost in Acute Respiratory Distress Syndrome (ThIlo) (NCT03111212). 

19. Treatment of ARDS With Sivelestat Sodium (TOAWSS) (NCT04909697). 

20. Pirfenidone to Prevent Fibrosis in Ards. (PIONEER) (NCT05075161). 

21. Birukova AA, Fu P, Xing J, Birukov KG. Rap1 mediates protective effects of iloprost against ventilator-induced lung injury. https://doi.org/101152/japplphysiol004622009. 2009;107: 1900–1910. doi:10.1152/JAPPLPHYSIOL.00462.2009

22. Sakashita A, Nishimura Y, Nishiuma T, Takenaka K, Kobayashi K, Kotani Y, et al. Neutrophil elastase inhibitor (sivelestat) attenuates subsequent ventilator-induced lung injury in mice. Eur J Pharmacol. 2007;571: 62–71. doi:10.1016/J.EJPHAR.2007.05.053

23. Li Y, Li H, Liu S, Pan P, Su X, Tan H, et al. Pirfenidone ameliorates lipopolysaccharide-induced pulmonary inflammation and fibrosis by blocking NLRP3 inflammasome activation. Mol Immunol. 2018;99: 134–144. doi:10.1016/J.MOLIMM.2018.05.003

24. Shah MD, Sumeh AS, Sheraz M, Kavitha MS, Maran BAV, Rodrigues KF. A mini-review on the impact of COVID 19 on vital organs. Biomed Pharmacother. 2021;143: 112158. doi:10.1016/J.BIOPHA.2021.112158

25. Ramasamy S, Subbian S. Critical Determinants of Cytokine Storm and Type I Interferon Response in COVID-19 Pathogenesis. Clin Microbiol Rev. 2021;34. doi:10.1128/CMR.00299-20

26. Middleton EA, Zimmerman GA. COVID-19–Associated Acute Respiratory Distress Syndrome: Lessons from Tissues and Cells. Crit Care Clin. 2021;37: 777. doi:10.1016/J.CCC.2021.05.004

27. Richard J-C, Yonis H, Bitker L, Roche S, Wallet F, Dupuis C, et al. Open-label randomized controlled trial of ultra-low tidal ventilation without extracorporeal circulation in patients with COVID-19 pneumonia and moderate to severe ARDS: study protocol for the VT4COVID trial. Trials. 2021;22: 692. doi:10.1186/S13063-021-05665-Z

28. Swenson KE, Swenson ER. Pathophysiology of Acute Respiratory Distress Syndrome and COVID-19 Lung Injury. Crit Care Clin. 2021;37: 749. doi:10.1016/J.CCC.2021.05.003

29. Matthay MA, Zemans RL, Zimmerman GA, Arabi YM, Beitler JR, Mercat A, et al. Acute respiratory distress syndrome. Nat Rev Dis Prim. 2019;5. doi:10.1038/S41572-019-0069-0

30. Nieri D, Neri T, Barbieri G, Moneta S, Morelli G, Mingardi D, et al. C-C motive chemokine ligand 2 and thromboinflammation in COVID-19-associated pneumonia: A retrospective study. Thromb Res. 2021;204: 88–94. doi:10.1016/J.THROMRES.2021.06.003

31. Sierra B, Pérez AB, Aguirre E, Bracho C, Valdés O, Jimenez N, et al. Association of Early Nasopharyngeal Immune Markers With COVID-19 Clinical Outcome: Predictive Value of CCL2/MCP-1. Open Forum Infect Dis. 2020;7: 1–5. doi:10.1093/OFID/OFAA407

32. Meizlish ML, Pine AB, Bishai JD, Goshua G, Nadelmann ER, Simonov M, et al. A neutrophil activation signature predicts critical illness and mortality in COVID-19. Blood Adv. 2021;5: 1164–1177. doi:10.1182/BLOODADVANCES.2020003568

33. Shaath H, Vishnubalaji R, Elkord E, Alajez NM. Single-Cell Transcriptome Analysis Highlights a Role for Neutrophils and Inflammatory Macrophages in the Pathogenesis of Severe COVID-19. Cells 2020, Vol 9, Page 2374. 2020;9: 2374. doi:10.3390/CELLS9112374

34. Li H, Xiang X, Ren H, Xu L, Zhao L, Chen X, et al. Serum Amyloid A is a biomarker of severe Coronavirus Disease and poor prognosis. J Infect. 2020;80: 646–655. doi:10.1016/J.JINF.2020.03.035

35. Gao S, Ding B, Lou W. microRNA-Dependent Modulation of Genes Contributes to ESR1’s Effect on ERα Positive Breast Cancer. Front Oncol. 2020;0: 753. doi:10.3389/FONC.2020.00753

36. Shen S, Kong J, Qiu Y, Yang X, Wang W, Yan L. Identification of core genes and outcomes in hepatocellular carcinoma by bioinformatics analysis. J Cell Biochem. 2019;120: 10069–10081. doi:10.1002/JCB.28290

37. Ma Q, Xu Y, Liao H, Cai Y, Xu L, Xiao D, et al. Identification and validation of key genes associated with non-small-cell lung cancer. J Cell Physiol. 2019;234: 22742–22752. doi:10.1002/JCP.28839

---

## [Decision Letter · Decision Letter 1]

10 Nov 2021

Core genes involved in the regulation of acute lung injury and their association with COVID-19 and tumor progression: a bioinformatics and experimental study

PONE-D-21-28077R1

Dear Dr. Sen'kova,

We’re pleased to inform you that your manuscript has been judged scientifically suitable for publication and will be formally accepted for publication once it meets all outstanding technical requirements.

Kind regards,

You-Yang Zhao

Academic Editor

PLOS ONE

Additional Editor Comments (optional):

Reviewers' comments:

Reviewer's Responses to Questions

**Comments to the Author**

1. If the authors have adequately addressed your comments raised in a previous round of review and you feel that this manuscript is now acceptable for publication, you may indicate that here to bypass the “Comments to the Author” section, enter your conflict of interest statement in the “Confidential to Editor” section, and submit your "Accept" recommendation.

Reviewer #1: All comments have been addressed

Reviewer #2: All comments have been addressed

2. Is the manuscript technically sound, and do the data support the conclusions?

Reviewer #1: Yes

Reviewer #2: Yes

3. Has the statistical analysis been performed appropriately and rigorously? 

Reviewer #1: Yes

Reviewer #2: Yes

4. Have the authors made all data underlying the findings in their manuscript fully available?

Reviewer #1: Yes

Reviewer #2: Yes

5. Is the manuscript presented in an intelligible fashion and written in standard English?

Reviewer #1: Yes

Reviewer #2: Yes

6. Review Comments to the Author

Reviewer #1: The author answered all the questions and revised the manuscipt. It is an really interesting study and provides some new methods to perform bioinformatic analysis of the published data, getting new ideas in the pathogenesis of ALI.

Reviewer #2: all comments have been addressed properly with new information. this manuscript is acceptable after the revision.

7. PLOS authors have the option to publish the peer review history of their article (what does this mean?). If published, this will include your full peer review and any attached files.

Reviewer #1: No

Reviewer #2: No

---

## [Editor Report · Acceptance letter]

12 Nov 2021

PONE-D-21-28077R1 

Core genes involved in the regulation of acute lung injury and their association with COVID-19 and tumor progression: a bioinformatics and experimental study 

Dear Dr. Sen’kova:

I'm pleased to inform you that your manuscript has been deemed suitable for publication in PLOS ONE. Congratulations! Your manuscript is now with our production department. 

Kind regards, 

on behalf of

Dr. You-Yang Zhao 

Academic Editor

PLOS ONE